# Nonvacuous Generalization Bounds For Deep Networks With Improved Size Dependence

## Abstract

Despite being massively overparameterized, deep neural networks exhibit a remarkable ability to generalize well to unseen data. Existing generalization bounds fail to explain this phenomenon, often becoming vacuous due to their strong dependence on network depth and width. To address this, we introduce novel nonvacuous generalization bounds for deep networks, offering tighter estimates of their Rademacher complexity by introducing a new analysis of covering number, which exhibits much milder depth dependence. Our bounds grow at a much slower rate of $O(\sqrt{Dpr})$, with network depth $D$, width $p$, and weights of rank $r$, compared to previous works that scale at a rate $O(\sqrt{D^3pr})$. Moreover, under certain plausible assumptions on the width of the network, we establish bounds that grow at a sublogarithmic rate of $O(\sqrt{log\ D})$ with depth. This novel bound is much tighter and represents a substantial improvement over prior bounds that scale at a polynomial rate with depth. We provide rigorous empirical validation, demonstrating that our bounds offer consistently tighter estimates compared to the state-of-the-art results. Thus, our bounds offer improved insight into the excellent generalization capabilities of deep overparameterized networks.

## 1 Introduction

Deep learning models have achieved remarkable success in many domains, including computer vision, Natural Language Processing, and so on. Yet, a fundamental theoretical puzzle remains: Why do over-parameterized deep networks, with many more parameters than training data, still generalize well to unseen data (Zhang et al., 2021)? Providing a theoretical justification for this generalization puzzle has remained elusive.

Generalization bounds are a powerful tool for assessing the generalization performance of deep neural networks. Traditional learning theory offers bounds, such as those based on the VC dimension (Vapnik & Chervonenkis, 2015) that are derived from the principle of uniform convergence (Nagarajan & Kolter, 2021). However, these bounds typically grow in proportion to the number of parameters in the network. For massive neural networks, these bounds become *vacuous* and cannot truly capture the generalization gap. Thus, these bounds are of little use in explaining the good generalization behavior of very large networks because of their explicit dependence on the size of these networks.

To address these limitations, researchers have explored alternative measures, such as the *norm* of the network weights (Neyshabur et al., 2015) and the classifier *margin* (Bartlett et al., 2017), to provide tighter bounds that are nonvacuous, more conservative, and correlate better with the generalization gap compared to those based on traditional methods.

Although bounds based on these measures have demonstrated a significant improvement, a new challenge has emerged: even these tighter bounds scale poorly with increasing depth and width of the network, whereas empirical observation suggests that the generalization gap of a network is almost independent of its width and grows slowly with its depth. For instance, Neyshabur et al. (2015) defines bounds, based on weight norms, that grow exponentially with the depth $D$ of the network (at a rate $O(2^D)$) and are thus vacuous, despite being tighter than traditional bounds based on the VC dimension. Similarly, Bartlett et al. (2017) and Neyshabur et al. (2018) suggest bounds that grow at a fast polynomial rate $O(\sqrt{D^3pr})$ with the network

dimensions, where $D$ denotes depth, $p$ the maximum number of hidden units in any layer, and $r$ denotes the rank of the weight matrices, even in special cases of bounded norms in each layer. However, these bounds still grow at a faster rate with the depth of the network, which might lead to loose bounds.

To address their shortcomings, we propose novel generalization bounds that exhibit a significantly milder dependence on the network dimensions. Our bounds grow only at a rate $O(\sqrt{Dpr})$, and in the special case of a bounded width of the network, the dependence becomes even more conservative, growing only sub-logarithmically with depth at a rate $O(\sqrt{log\ D})$ and having an overall growth rate of $O(pr + \sqrt{log(Dpr)})$. Thus, our bounds are much tighter, nonvacuous, growing at a slower pace with increasing network depth, and thus provide a better theoretical explanation for the good generalization performance of over-parameterized deep networks.

To provide a detailed derivation of the bounds and show that it captures the generalization gap well, we have organized the paper as follows. In Section 2, we present related works that attempted to provide nonvacuous generalization bounds for deep networks to provide context for our work. In Section 3, we provide a rigorous derivation of our generalization bounds, starting with the general condition and then proving it for the special case. We then compare our bounds with related bounds on the generalization gap of deep networks. In Section 4, we experimentally evaluate our bounds and show that they are tighter than the related bounds suggested previously. In Section 5, we provide a detailed discussion of the empirical comparison of the different bounds evaluated on networks with different depths. Finally, we conclude in Section 6 while suggesting interesting directions for future research in Section 7.

In general, our work makes the following contributions:

- We propose a novel generalization bound that is nonvacuous and has a better dependence on the network dimensions (depth and width) than previously suggested bounds. This marks a step forward in obtaining tighter bounds that scale well with the network size.

- Under certain realistic and practical assumptions, we provide bounds that scale only sub-logarithmically with a rate $O(\sqrt{log\ D})$ with depth $D$ for a fixed maximum width $p$, which is slower than the prior works that scale at a rate of at least $O(\sqrt{D})$. Crucially, our assumptions are more broadly applicable and more plausible than the restrictive ones made in works such as (Golowich et al., 2018).

- Through extensive experiments, we provide empirical evidence demonstrating the tightness of our bound and its superior dependence on depth. By demonstrating the significantly lower values yielded by our bound over networks of increasing depth compared to competitive baselines, we confirm that our nonvacuous bounds are much tighter and scale better with depth, providing a superior alternative to the prior bounds.

## 1.1 Preliminaries

This section defines key concepts related to DNNs, loss functions, and generalization bounds. The key terms defined here will be essential for the subsequent analysis.

Given an input $x$, we denote the output of a $D$ layer neural network as $f(x) = f_{w_d}(f_{w_{d-1}}(..f_{w_1}(x)))$, where $w_d, w_{d-1}, .., w_1$ denote the weights of this network in layers $d, d-1, ..., 1$. The output of any intermediate layer $d$ is given as $f_{w_d}(y_{d-1}) = \sigma(W_d y_{d-1})$, where $\sigma(.)$ is the activation function that, in our case, is the ReLU activation (Agarap, 2019) and $y_d$ denotes the output of the $d_{th}$ layer.

We denote loss functions as $g(f(x), y)$ where the first argument specifies the network output corresponding to the input $x$, and the second specifies the target $y$. We then denote a class of loss functions defined over the function class $\mathcal{F}_D$ representing $D$-layer neural networks as

$$\mathcal{G}(\mathcal{F}_\mathcal{D}) = \{g(f(w, x_i), y_i) : w \in \mathcal{W}, (x_i, y_i) \in (\mathcal{X}, \mathcal{Y})\} \tag{1}$$

where $\mathcal{W}$ specifies the set representing all plausible values of the network weights and $\mathcal{X}, \mathcal{Y}$ denote the sets of possible inputs and their corresponding observations, respectively. Given a set of $m$ inputs

$(x_1, y_1), (x_2, y_2), ..(x_m, y_m)$, the generalization error of a classifier $f$ is the difference between its true (or expected) risk and its empirical risk on this subset of size $m$ and can be written as:

$$\mathbb{P}[arg \max_i (f(x)_i) \neq y] - \frac{1}{m} \sum_{i=1} g(f(x_i), y_i) \qquad (2)$$

Let us take the task of classification as an example to elucidate this concept further. Specifically, we consider the multiclass classification with $c$ classes. Consider a particular bounded form of the loss function $g(.)$, i.e., the $margin - loss$ or $ramp - risk$. For a given input $x$ with the correct class $y \in \{1, 2, .., c\}$, let us define $m_W^{x,y} = f_y(x) - \max_{i \neq y} f_i(x)$. We can then define the class of $margin\ loss$ functions with a margin $\gamma$ as:

$$\mathcal{G}_\gamma(\mathcal{F}_D) = \{g_\gamma(f(w,x), y) : \forall x \in \mathcal{X}, \forall y \in \mathcal{Y}, \forall f \in \mathcal{F}_D, \forall w \in \mathcal{W}\} \qquad (3)$$

where

$$g_\gamma(f(x), y) = \begin{cases} 0, & m_W^{x,y} > \gamma \\ 1 - \frac{m_W^{x,y}}{\gamma}, & 0 \leq m_W^{x,y} \leq \gamma \\ 1, & m_W^{x,y} < 0 \end{cases} \qquad (4)$$

Thus, the margin-loss being 1 indicates an incorrect prediction, while it is 0 when the model output for the correct class is greater than that of all other classes by the margin $\gamma$. Otherwise, if the output corresponding to the correct class is greater than that of the maximum of other classes by a value $\leq \gamma$, this loss goes down from 1 towards 0 as this difference reaches $\gamma$.

After defining these crucial concepts, we now define some important terms for understanding generalization bounds. The empirical Rademacher complexity (Bartlett & Mendelson, 2002) $\mathfrak{R}_n(\mathcal{G}_\gamma(\mathcal{F}_D))$ for a sample set of size $n$ over a function class $\mathcal{F}_D$ is described as

$$\mathfrak{R}_n(\mathcal{G}_\gamma(\mathcal{F}_D)) = \mathbb{E}_{\epsilon_i \in [-1, +1]^n} [\sup_{f \in \mathcal{F}_D} |\frac{1}{n} \sum_{i=1}^n \epsilon_i g(f(x_i), y_i)|] \qquad (5)$$

where $\epsilon$ is a vector that contains -1 or +1 with equal probability. This quantity determines the richness and complexity of a function class on data drawn from a distribution $(\mathcal{X}, \mathcal{Y})$ and is related to the complexity of the network. Intuitively, the Rademachar complexity determines the network's ability to fit random labels for a given dataset. The more complex a function class is, the greater will be its potential to fit random samples with noise. This quantity will play a crucial role in the derivation of our generalization bounds.

Next, we introduce the concept of $covering\ number$ which will be useful in the analysis of the generalization of deep networks and is connected to the Rademacher complexity defined above. The covering number $\mathcal{N}(\mathcal{F}, \epsilon, \|.\|_F)$ of a function class $\mathcal{F}$ output by a neural network is defined as the number of balls of radius $\epsilon$, in the $\|.\|_F$ norm, that can tightly cover the space of functions $\mathcal{F}$.

To understand it more intuitively, imagine a space of all functions that your neural network can represent (with a possible set $\mathcal{W}$ of weights) and suppose you want to approximate any function with up to $\epsilon$ precision. To achieve this, you start placing balls of radius $\epsilon$ in this space so that each function overlaps with at least one ball. The number of balls placed to achieve this is the covering number $\mathcal{N}(\mathcal{F}, \epsilon, \|.\|_F)$ for the given function space.

Bartlett et al. (2017) connects the generalization gap to the Rademachar complexity as follows: Given a function class representing $D$-layer neural networks $\mathcal{F}_D$, a sample set $S$ of size $n$, margin loss function $g_\gamma$, and a real number $\delta \in [0, 1]$, we have, with probability $> 1 - \delta$, that the generalization error is bounded with respect to the Rademacher complexity via the following inequality:

$$\mathbb{P}[argmax_i (f(x)_i) \neq y] \leq \frac{1}{n} \sum_{i=1}^n g_\gamma(f(x_i), y_i) + 2\mathfrak{R}_n(\mathcal{G}_\gamma(\mathcal{F}_D)) + 3\sqrt{\frac{ln(2/\delta)}{n}} \qquad (6)$$

The generalization bound of a predictor can also be derived using the PAC-Bayesian framework (McAllester, 2003), which takes into account the full posterior distribution over the weights of the neural network to derive upper bounds on generalization error. According to this framework, we first define a prior distribution of the weights $w$ denoted as $P$ that encodes our *prior beliefs* about the network and is independent of training data. Now, suppose a predictor $f_\mathbf{w}$ (not necessarily a neural network) with trained weights $\mathbf{w}$ that is drawn from a posterior distribution $Q$ and depend on the training data. Then, according to the PAC-Bayesian framework, with a probability $> 1 - \delta$ over a random draw of the training data of size $m$, we have

$$\mathbb{E}[L(f_\mathbf{w})] \leq \mathbb{E}[\widehat{L}(f_\mathbf{w})] + \sqrt{\frac{KL(\mathbf{w}\|P) + ln(2m/\delta)}{m-1}} \tag{7}$$

where $L(f_\mathbf{w})$ is the actual or expected risk and $\widehat{L}(f_\mathbf{w})$ is the empirical risk over the dataset. Thus, the PAC-Bayesian framework bounds the generalization error using the KL divergence between the posterior distributions $Q$ of the trained weights and the prior weights $P$. Thus, the more divergent the distribution of the trained weights and the prior weights, the higher the generalization gap. The bounds derived using the PAC-Bayesian principles have also been connected to the *sharpness* of the network and the empirical observation that predictors with *flat minima* in the weight space often generalize well (Keskar et al., 2017). This is because on flat minima the set of solutions $\mathbf{w}$ is more uniformly distributed in a large volume around the prior distribution $P$ and thus the KL term in equation (7) is smaller. Hence, small perturbations of the weights do not lead to a large change in empirical loss, predictor output $f_\mathbf{w}$, and the KL term. Hence, in such cases, a perturbation analysis of the predictor by introducing small random noise to obtain $f_{\mathbf{w+u}}$ and observing its effect on the change in the predictor output ($|f_{\mathbf{w+u}} - f_\mathbf{w}|$) would allow us to draw generalization bounds. We will use this perturbation technique to derive our generalization bound in the general case.

## 1.2 Notations

Now we state some important notations that would be regularly used in the mathematical expressions below. We will usually use $D, p$, and $r$ to denote the depth, maximum width, and the rank of the weight matrices of a neural network. Given a matrix $W_d$, which represents the weights of the $d_{th}$ layer of a neural network, we denote its spectral norm as $B_{d,2}$ or $\|W_d\|_2$, the Frobenius norm as $B_{d,F}$ or $\|W_d\|_F$, and $B_{d,2\to1}$ or $\|W_d\|_{2,1}$ denotes the 2,1-norm of the $d_{th}$ layer, which is described mathematically as $\|W\|_{2,1} = \sum_{i=0}^n \sqrt{\sum_{j=1}^m W_{ij}^2}$. This is just the sum of the $l_2$ norm of each row of the matrix $W_d$. For a given neural network, $J_{W_d}$ will denote the Jacobian of the $d_{th}$ layer, $B_{1:d}^{jac}$ denotes an upper bound on the Jacobian norm of the first $d$ layers with respect to the input and $B_{\backslash d}^{jac}$ denotes the Jacobian upper bound of the full network excluding the $d_{th}$ layer. Generally, $B_{i:d}^{jac}$ denotes the upper bound on the Jacobian of the sub-network starting at the $i_{th}$ layer and ending at the $d_{th}$. Furthermore, $\gamma$ denotes the classifier margin. $R$ will denote the $L_2$ norm of the input matrices $\|X\|_2$, i.e., $R = \|X\|_2 = \sqrt{\sum_{i=1}^m \|x_i\|_2^2}$. $\rho_1, \rho_2, .., \rho_D$ denotes the Lipschitz constant of the respective layers in a $D$-layer neural network and $\mathcal{N}(\mathcal{F}, \epsilon, \|.\|_p)$ denotes the covering number of the function class $\mathcal{F}$ learnable by the neural network. This covering number determines the number of balls of size $\epsilon$ required to cover the whole class of neural network functions represented by $\mathcal{F}$, with the distance measured in the $\|.\|_p$ norm.

## 2 Related Works

### 2.1 Nonvacuous generalization bounds for deep networks

Traditional generalization bounds for neural networks, such as those based on the VC dimension (Vapnik & Chervonenkis, 2015) scale linearly with the number of network parameters. Although theoretically grounded in statistical learning theory, these bounds are vacuous for overparameterized neural networks, thus providing little insight into the generalization performance of deep networks. A recent line of work has attempted to address this limitation by proposing nonvacuous generalization bounds that are some functions of the *norm* of the weights (Bartlett et al., 2017; Neyshabur et al., 2018; 2015) and the classifier *margin*. Unlike traditional bounds, these methods offer more favorable scaling and better estimation of the generalization gap.

Early works in this direction (Neyshabur et al., 2015) proposed bounds that are explicitly based on the $l_{p,q}$ norm of the network weights. In this approach, the $l_p$ norm of the weights incoming to each hidden unit is first computed, which is then followed by a $l_q$ norm of the resulting vector. While this bound is a notable improvement over those based on the VC dimension, it still suffers from a critical limitation. This bound scales exponentially with the depth of the network (scaling at a rate $O(\frac{2^D \prod_d B_{d,F}}{\gamma\sqrt{m}})$), where $D$ is the depth), which does not align well with empirical observations according to which the generalization gap grows slowly and often improves with depth (Nakkiran et al., 2020).

One of the seminal works that addressed the fast scaling of the previous bounds with depth is the one proposed by Bartlett et al. (2017). This bound leverages the concept of covering number for neural networks to derive nonvacuous bounds that are proportional to the $l_{2,3}$ norm of the ratio of the Frobenius norm to the spectral norm of the weights across the layers to provide tighter generalization estimates.

$$\frac{\prod_{d=1}^{D} B_{d,2} \log p}{\gamma\sqrt{m}} \sqrt{\left( \sum_{d=1}^{D} \frac{B_{d,2\to1}^{2/3}}{B_d^{2/3}} \right)^3} \tag{8}$$

However, their bound grows at a rate of at least $O(\frac{\sqrt{D^3 pr}}{\sqrt{m}})$ (assuming that the ratio of spectral norm to the margin $\frac{\prod_{d=1}^{D} B_{d,2} \log p}{\gamma\sqrt{m}}$ stays bounded by a constant) which is proportional to the square root of the cube of the network depth. Although their bounds showed an improved growth rate with network dimensions compared to (Neyshabur et al., 2015), it still did not result in bounds with a tighter dependence on depth.

Neyshabur et al. (2018) later proposed a tighter bound that is based on PAC-Bayesian principles (McAllester, 2003) and is proportional to the sum of the ratio of the squares of the Frobenius norm to the spectral norm across all layers multiplied by $D^2 p$.

$$\frac{\prod_{d=1}^{D} B_{d,2} \, \log(Dp)}{\gamma\sqrt{m}} \sqrt{D^2 p \left( \sum_{d=1}^{D} \frac{B_{d,F}^2}{B_{d,2}^2} \right)} \tag{9}$$

Despite being an improvement over the previous bound, this bound still grows at the rate $\frac{\sqrt{D^3 pr}}{\sqrt{m}}$ due to the extra $D^2 p$ term, and thus scales at the same rate as the previous bound, since the sum of the ratio of the Frobenius norm to the spectral norm scales as $O(D)$. Due to the fast growth rate of these bounds with network depth, which contradicts empirical findings, these bounds cannot provide an accurate estimate of the generalization gap, especially for very deep networks.

The fast growth rates of the previous bounds present a significant challenge in providing tight and nonvacuous estimates of the generalization gap of the network. This can be a significant hurdle in providing an estimate that is closer to the actual generalization gap of very deep networks, as these bounds grow very fast with network depths, while empirical observation suggests that the generalization error grows more slowly with depth (slower than a polynomial rate), and even adding depth under certain conditions improves generalization (Nakkiran et al., 2020; Advani & Saxe, 2017).

To address the limitations of previous bounds and provide better depth scaling, Li et al. (2019) provided a tighter bound that scales as $O(\frac{\sqrt{Dpr * \log C_{net}}}{\gamma\sqrt{m}})$, where $C_{net}$ is a term that is proportional to the square root depth of the network and its Jacobian. Thus, their bound scale as $O(\sqrt{Dpr * \log(c\sqrt{D})})$, where $c$ is a constant, which is tighter compared to the previous bounds. A more recent attempt at obtaining size-independent bounds was made in (Golowich et al., 2018), where the authors propose a bound which is proportional to the logarithm of the ratio of the product of the Frobenius norm of the layer weights and its lower bound $\Gamma \leq \prod_{d=1}^{D} B_{d,F}$:

$$\prod_{d=1}^{D} \frac{B_{d,F}}{\gamma} \, \min\left\{ \frac{1}{\sqrt[4]{m}} \sqrt{\log \frac{\prod_{d=1}^{D} B_{d,F}}{\Gamma}}, \sqrt{\frac{D}{m}} \right\} \tag{10}$$

However, their bounds only hold in a very stringent condition where the Frobenius norms of the weights in each layer $d$ are upper-bounded by some constant $B_{d,F}$, while their product is lower-bounded by $\Gamma$. Moreover, they used the product of the Frobenius norm of the layers, instead of the tighter spectral norm used by prior bounds, which can lead to terms that scale exponentially with depth ($O(r^D)$) in the worst case, resulting in loose bounds. Lastly, their bound is independent only in the case where the first term in the min operation ($\sqrt{\frac{1}{\sqrt[4]{m}} log \frac{\prod_{d=1}^{D} B_{d,F}}{\Gamma}}$) is sufficiently smaller than the second term $\sqrt{D/m}$, which is more rarely satisfied due to an almost exponential growth of $B_{d,F}$ with depth.

Compared to previous work, we propose bounds that exhibit a significantly better dependence on network dimensions (both depth and width). Our bounds scale as $O(\frac{\sqrt{Dpr}}{\sqrt{m}})$, and thus have a better depth scaling than all the previous bounds, including (Li et al., 2019) which scaled as $O(\sqrt{\frac{Dpr*log(D)}{m}})$.

Moreover, under certain practical assumptions on the width of network layers, our bounds achieve a remarkable sub-logarithmic scaling with depth at a rate $\left(\frac{\sqrt{log(D)}}{\sqrt{m}}\right)$ for a constant maximum width $p$. This provides a significantly tighter and realistic estimate of the generalization gap of deeper neural networks with many more layers, addressing a major limitation of previous work and aligning well with the empirical observation.

## 3  An improved generalization bound with better size dependence

This section presents our novel generalization bounds and provides rigorous mathematical derivations for them. We begin by establishing our bound in the general case without any restrictions on the network norm or its size by using the PAC-Bayesian principles (McAllester, 2003) via a perturbation bound on the network weights. We then demonstrate that under certain practical constraints on the width of the network layers, a tighter generalization bound can be obtained using a covering number argument that bounds the Rademacher complexity of the network. This tighter bound scales only sub-logarithmically with the depth $D$ of the network and has a tighter dependence on both depth and width.

In deriving the growth rates for the bounds, we assume that the spectral norm $\|W_i\|_2$ of each individual layer is equal to 1 (i.e., $\|W_i\|_2 = 1$), which will prevent the product of spectral norms from growing with depth. This is a common assumption used in different papers, including (Li et al., 2019; Neyshabur et al., 2018), when deriving growth rates of generalization bounds. Moreover, we exclude the factor $\sqrt{ln(Dp)}$ that appears in our first bound derived in Theorem 2 when evaluating the growth rates of this bound and is also common with the bound in (Neyshabur et al., 2018).

### 3.1  The bound in the general case

To derive this bound, we first introduce the following PAC-Bayesian bound based on the upper bound on the KL divergence between the distribution of the trained weights and the weights drawn from a prior distribution $P$ across all the $D$ layers, which will serve as a key component in our analysis.

**Lemma 1.** *(Neyshabur et al., 2018) Let $f_{\mathbf{w}}(x) : \mathcal{X} \to \mathbb{R}^k$ be any neural network with parameters $\mathbf{w}$ and $P$ be a prior distribution on its weights that is independent of the training data. Then for any $\gamma, \delta > 0$, with probability $> 1 - \delta$ over training set of size $m$, for any $\mathbf{w}$ and any random perturbation $\mathbf{u}$ s.t. $\mathbb{P}_{\mathbf{u}}[max_{x \in \mathcal{X}} |f_{\mathbf{w}+\mathbf{u}}(x) - f_{\mathbf{w}}(x)| < \frac{\gamma}{4}] > \frac{1}{2}$, we have*

$$\mathbb{P}[arg \max_i f(x)_i \neq y] < \frac{1}{n} \sum_{i=1}^{n} g_\gamma(f(x_i), y_i) + 4\sqrt{\frac{KL((\mathbf{w}+\mathbf{u})\|P)}{m-1}} \tag{11}$$

*where $\gamma$ denotes the margin of the network, $P$ denotes the prior distribution of weights, $KL(Q\|P)$ denotes the KL divergence between the distributions $Q$ and $P$, $g_\gamma(.)$ denotes the margin loss function.*

This KL bound serves as an essential component for bounding the complexity of the network and is based on the notion of *sharpness* of the network or the sensitivity of the empirical risk to small weight perturbation, which will be useful in deriving our generalization bounds.

To derive our bounds, we deviate from previous work (Neyshabur et al., 2018) by replacing the prior distribution $P$ with a Normal distribution $\mathcal{N}(W_0, \sigma^2)$, with its mean representing the weights at initialization $W_{i_0}$, to obtain tighter bounds. This choice was informed by the empirical observation that the weights of the neural network tend to remain closer to their initialization throughout the training phase and is also theoretically explained in (Nagarajan & Kolter, 2019). Based on this, we propose our novel margin-based generalization bounds in the theorem below.

**Theorem 2.** *Let us assume a D-layer neural network with weights in each layer given by $W_1, W_2, ...W_i, .., W_D$ and these weights during initialization are given as $W_{1_0}, W_{2_0}, W_{3_0}, .., W_{i_0}, ..W_{D_0}$. If $(x_1, y_1), (x_2, y_2), (x_3, y_3), ...., (x_m, y_m)$ are drawn iid from a distribution, then with probability at least $1 - \delta$, assuming the prior distribution $P$ on weights is a normal distribution with mean 0, for any $R, p, D$, the true error rate is bounded by :*

$$\mathbb{P}[\arg\max_i f(x)_i \neq y] \leq \hat{\mathcal{R}}(f) + \sqrt{\frac{\frac{R^2 \prod_{i=1}^D \|W_i\|_2^2 \, p \, ln(4pD)}{\gamma^2} \sum_{i=1}^D \frac{\|W_i\|_F^2}{\|W_i\|_2^2} + \frac{p}{D} + Dp(2ln(D) - 1) + ln(Dm)/\delta}{m}} \tag{12}$$

*Furthermore, if we instead assume that the prior distribution of weights in each layer is a Normal distribution with mean $W_{i_0}$, then assuming the same conditions as before, with probability $> 1 - \delta$, the true error rate is bounded by*

$$\mathbb{P}[\arg\max_i f(x)_i \neq y] \leq \hat{\mathcal{R}}(f) + \sqrt{\frac{\frac{R^2 \prod_{i=1}^D \|W_i\|_2^2 \, p \, ln(4pD)}{\gamma^2} \sum_{i=1}^D \frac{\|W_i - W_{i_0}\|_F^2}{\|W_i\|_2^2} + \frac{p}{D} + Dp(2ln(D) - 1) + ln(Dm)/\delta}{m}} \tag{13}$$

*where $\gamma$ denotes the margin of the network, $\hat{\mathcal{R}}(f) = m^{-1} \sum_{i=1}^m \mathbb{I}[f(x_i)_{y_i} \leq \max_{j \neq y_i} f(x_i)_j + \gamma]$ is the empirical risk and $R = \max_x \|x\|_2$.*

The full derivation of this theorem is described in the Appendix. The above theorem states our generalization bounds for the general condition. The first equation (12) of our theorem scales with the Frobenius norm of the network weights, whereas the second equation (13) depends on the distance of the weights from the initialization. Since the weights travel a small distance from their initialization, as observed empirically, we will demonstrate that (13) provides a tighter bound on the generalization error than (12). Note that our PAC-Bayesian bound grows with the order $O(\sqrt{Dpr})$, (assuming $\|W_i\|_2$ satisfies $\|W_i\|_2 = 1$ for all the layers $i$) which is much tighter than the bound proposed in (Bartlett et al., 2017) which is based on the following upper bound on the Rademacher complexity of the network:

$$\mathfrak{R}_n(\mathcal{F}_D) \overset{(i)}{\leq} \inf_{\alpha > 0}\left(\frac{4\alpha}{\sqrt{n}} + \frac{12}{\sqrt{n}} \int_\alpha^{\sqrt{n}} \sqrt{ln \, \mathcal{N}(\mathcal{F}, \epsilon_i, \|.\|_\infty)}\right) \leq \frac{8}{n} + \frac{\prod_{i=1}^D \rho_i \|W_i\|_2 ln(n) ln(2W)}{\gamma n} \sqrt{\left(\sum_{i=1}^D \left(\frac{\|W_i\|_{2,1}^{2/3}}{\|W_i\|_2^{2/3}}\right)\right)^3} \tag{14}$$

where inequality $(i)$ is a minor variant of the Dudley integral entropy (Bartlett et al., 2017) bound on the Rademacher complexity, and this overall bound grows at a rate $O(\sqrt{D^3 pr})$ under the same assumptions on spectral norms of layers. Moreover, incorporating the distance of the weights from initialization in our measures leads to tighter bounds that align well with the empirical observation suggesting that the weights stay consistently closer to their initialization throughout training (Nagarajan & Kolter, 2019).

## 3.2 A tighter bound with logarithmic depth dependency

Now, we derive our bound for the special case where we assume that the number of hidden units in any layer is bounded by $p$ but is sufficiently large ($p > 512$, which is a common upper bound in many overparameterized deep networks). Under this assumption, we will use Jensen's inequality to obtain a tighter bound, which only scales sub-logarithmically with depth $D$.

To derive this bound, we leverage the following Dudley entropy integral bound from statistical learning theory that bounds the Rademacher complexity (Liao, 2020) $\mathfrak{R}_n(\mathcal{F})$ of a function class $\mathcal{F}$ using the covering number:

$$\mathfrak{R}_n(\mathcal{F}) \leq \inf_{\beta > 0} \left( 4\beta + \frac{12}{\sqrt{m}} \int_{\beta}^{B/2} \sqrt{log\, \mathcal{N}(\mathcal{F}, \epsilon, L_2(\mathcal{F}))} \right) \tag{15}$$

where $\mathcal{N}(\mathcal{F}, \epsilon, L_2(\mathcal{F}))$ denotes the $\epsilon$-covering of the neural network function class $\mathcal{F}$ with respect to the metric $L_2(\mathcal{F})$ in the function class $\mathcal{F}$ and $m$ is the sample set size. Now, based on the assumptions that our network is of finite width $p$, we derive an upper bound on the Rademacher complexity in the following theorem.

**Theorem 3.** *Let us assume a $D$-layer neural network, with at most $p$ hidden units in any layer, which is sufficiently large. We assume that the weights in each layer have a maximum rank of $r$ and are denoted by $W_1, W_2, ... W_i, .., W_D$.*

*Let $S = (x_1, y_1), (x_2, y_2), (x_3, y_3), ...., (x_m, y_m)$ be a set of $m$ training examples drawn iid from a distribution. Let $\mathcal{F}_D$ denote the function class of all such neural networks. Then, with probability at least $1 - \delta$ over a random draw of the set $S$, the Rademacher complexity of the margin loss function class $\mathcal{G}_\gamma$ over the network function class $\mathcal{F}_D$ is bounded as*

$$\mathfrak{R}_n(\mathcal{G}_\gamma(\mathcal{F}_D)) = B_{1:D}^{jac} R \left( 16\sqrt{2} pr + \frac{6}{\gamma\sqrt{m}} \sqrt{log\left( 2Dpr + \sum_{i=1}^{D} \frac{\|W_i\|_F}{\|W_i\|_2 \sqrt{2}} \right) - log\, \gamma} \right) \tag{16}$$

*where $\gamma$ is the network margin, $R = \max_{x} \|x\|_2$, and $B_{1:D}^{jac}$ denotes the upper bound on the Jacobian of the entire network.*

The detailed proof of this theorem is relegated to the Appendix.

To derive the growth rate of this bound, we first assume that $\|W_i\|_2 = 1$ for all $i$. In addition, we utilize the fact that the spectral norm bounds the Frobenius norm via the inequality $\|W_i\|_F \leq \sqrt{r}\|W_i\|_2$. This holds since the Frobenius norm is the sum of singular values $\|W\|_F = \sqrt{\sum_{i=1}^{r} \sigma_i^2}$ and the spectral norm is the largest singular value $\sigma_1$. Hence, $\|W\|_F = \sqrt{\sum_{i=1}^{r} \sigma_i^2} \leq \sqrt{\sum_{i=1}^{r} \sigma_1^2} \leq \sqrt{r\sigma_1^2} = \sqrt{r}\sigma_1 = \sqrt{r}\|W\|_2$ Thus, using the inequality $\|W_i\|_F \leq \sqrt{r}\|W_i\|_2$ where $r$ is the rank of $W_i$, we get

$$\frac{\|W_i\|_F}{\|W_i\|_2} \leq \sqrt{r} \tag{17}$$

Thus, the individual terms within the summation in the log in Theorem 3 scale at the rate $O(2Dpr + D\sqrt{r}/\sqrt{2}) \approx O(2Dpr + D\sqrt{r/2})$, hence the value of the summation within the log in (16) can grow at a maximum rate of $O(Dpr)$ and the overall Rademachar complexity term derived in this theorem scales at a rate $O(pr + \sqrt{log\, Dpr}) \approx O(pr + \sqrt{log\, D + log\, pr})$ with depth $D$, maximum width $p$ and rank $r$. Thus, our bound scales only sub-logarithmically with respect to depth $D$.

Since, we assume that $\forall_i \|W_i\|_2 = 1$, the term $B_{1:D}^{jac}$ becomes equal to 1 and the generalization bound should grow at a rate $O(pr + \sqrt{log\, D + log\, pr}) \leq O(pr + \sqrt{log\, D} + \sqrt{log\, pr}))$ with respect to the network dimensions. Hence, on the basis of this observation, our proposed bound should grow only sub-logarithmically with depth $D$ for a fixed maximum width $p$ at a rate $O(\sqrt{log\, D})$. Now, substituting (16) into the following generalization bound (Bartlett et al., 2017) based on the Rademacher complexity

$$\mathbb{P}[arg\, \max_{i} (f(x)_i) \neq y_i)] \leq \hat{\mathcal{R}}(f) + 2\mathfrak{R}_n(\mathcal{F}) + 3\sqrt{\frac{ln(1/\delta)}{n}} \tag{18}$$

we finally get our refined generalization bound.

Table 1: Comparison of existing norm-based generalization bounds with our results for neural networks of depth $D$, $p$ hidden units in each layer, and weight matrices of rank $r$. $B_{d,F}, B_{d,2}$, and $B_{d,2\to1}$ denote the $\|W_d\|_F, \|W_d\|_2, \|W_d - W_{d_0}\|_{2,1}$ norms and $\Gamma \leq \prod_{d=1}^{D} \|W_d\|_2$ denotes the lower bound on product of Frobenius norm (Golowich et al., 2018). We also derive each bound under the constrain $\|W_d\|_2 = 1$ for all the layers $d = 1, 2, .., D$ which makes the common factor $\prod_{d=1}^{D} B_{d,2} = 1$.

| capacity bound | Original results | $\|W_d\|_2 = 1$ |
|---|---|---|
| (Neyshabur et al., 2015) | $\mathcal{O}\left(\frac{2^D \prod_{i=1}^{D} B_{d,F}}{\gamma\sqrt{m}}\right)$ | $\widetilde{\mathcal{O}}\left(\frac{2^D r^{D/2}}{\gamma\sqrt{m}}\right)$ |
| (Bartlett et al., 2017) | $\mathcal{O}\left(\frac{\prod_{d=1}^{D} B_{d,2} R \, log \, p}{\gamma\sqrt{m}}\left(\sum_{l=1}^{D} \frac{B_{d,2\to1}^{2/3}}{B_d^{2/3}}\right)^{3/2}\right)$ | $\widetilde{\mathcal{O}}(\frac{\sqrt{D^3 pr}}{\gamma\sqrt{m}})$ |
| (Neyshabur et al., 2018) | $\mathcal{O}\left(\frac{\prod_{d=1}^{D} B_{d,2} R}{\gamma\sqrt{m}}\sqrt{D^2 p \sum_{d=1}^{D} \frac{B_{d,F}^2}{B_{d,2}^2} \, log(Dp)}\right)$ | $\widetilde{\mathcal{O}}(\sqrt{\frac{D^3 pr}{\gamma\sqrt{m}}})$ |
| (Golowich et al., 2018) | $\mathcal{O}\left(\prod_{d=1}^{D} \frac{B_{d,F}}{\gamma} \, min\left\{\frac{\sqrt{log\frac{\prod_{d=1}^{D} B_{d,F}}{\Gamma}}}{m^{1/4}}, \sqrt{\frac{D}{m}}\right\}\right)$ | $\mathcal{O}\left(\frac{\sqrt{r^D} D}{\gamma\sqrt[4]{m}}\right)$ |
| (Li et al., 2019) | $\mathcal{O}\left(\frac{B_{1:d}^{jac}\sqrt{Dpr} \, log\left(\frac{B_{\backslash d,2}^{jac}\sqrt{Dm/r} \, max_x B_{d,2}}{sup \, g(f(\mathcal{W}_D, x))}\right)}{\gamma\sqrt{m}}\right)$ | $\widetilde{\mathcal{O}}\left(\frac{\sqrt{Dpr * log(D)}}{\gamma\sqrt{m}}\right)$ |
| Our bound | $\mathcal{O}\left(\frac{\prod_{d=1}^{D} B_{d,2} R}{\gamma\sqrt{m}}\sqrt{\sum_{d=1}^{D} \frac{B_{d,F}^2}{B_{d,2}^2} \, plog(Dp)}\right)$ | $\widetilde{\mathcal{O}}\left(\frac{\sqrt{Dpr}}{\gamma\sqrt{m}}\right)$ |
| Our bound (specialized) | $\mathcal{O}\left(\prod_{d=1}^{D} B_{d,2}\left(\frac{R}{\gamma\sqrt{m}}\sqrt{log\left(2Dpr + \sum_{d=1}^{D} \frac{B_{d,F}}{B_{d,2}\sqrt{2}}\right)} + pr\right)\right)$ | $\widetilde{\mathcal{O}}(\frac{\sqrt{log(D)+log(pr)}}{\gamma\sqrt{m}} + pr)$ |

**Theorem 4.** *Let's assume a D-layer neural network, with a maximum of p hidden units across all layers, which is sufficiently large ($p > 512$), weights in each layer given by $W_1, W_2, ...W_i, .., W_D$ and having a maximum rank $r$. If $(x_1, y_1), (x_2, y_2), (x_3, y_3), ...., (x_n, y_n)$ is a sample S of size n drawn iid from a distribution and assuming that the norm of each hidden unit's weight is bounded above by 1, then with probability at least $1 - \delta$, over random draw of S, the generalization error is bounded as*

$$\mathbb{P}[\underset{i}{argmax}(f(x)_i) \neq y] \leq \hat{\mathcal{R}}(f) + O\left(B_{1:d}^{jac}\left(\frac{\|X\|_2}{\gamma n}\sqrt{log\left(2Dpr + \sum_{i=1}^{D} \frac{\|W_i\|_F}{\|W_i\|_2\sqrt{2}}\right)} + Cpr\right) + \sqrt{\frac{ln(2/\delta)}{n}}\right) \quad (19)$$

*where $\gamma$ denotes the classifier margin, $\hat{R}(f) = \sum_{i=1}^{n} \frac{(argmax_j f(x_i)_j \neq y_i)}{n}$ is the empirical risk and $\|X\|_2 = \sqrt{\sum \|x_i\|_2}$.*

This states our bound in the specialized case. Note that our bound is tighter than Bartlett et al. (2017) as well as the more recent bound Li et al. (2019) for large depth, and scales only sub-logarithmically with depth. We now provide an empirical comparison of the growth rate and the tightness of our bounds compared to the other related bounds.

## 4 Experimental Evaluation

We evaluated and compared our generalization bound against the ones proposed in Bartlett et al. (2017); Neyshabur et al. (2018); Li et al. (2019)and Golowich et al. (2018) on the VGG-16 network and its deeper variants (VGG-19, VGG-21, and VGG-26) along with the ResNet-18 and 34 models (He et al., 2015) on the CIFAR-10 dataset (Krizhevsky, 2009). VGG-21 and VGG-26 models are extensions of VGG-19, where VGG-21 had one more convolutional layer in the last two blocks, each consisting of 512 filter layers. Similarly, VGG-26 was an extension of VGG-19 and had 3 more layers in the last two convolutional blocks with 512

filter layers in each block, and 1 more convolutional layer in the third block consisting of 256 filter layers. Furthermore, we also derived the bounds for the Residual networks, on the more complex CIFAR-100 and Tiny Imagenet dataset (Le & Yang, 2015) to validate the generalizability of our bound to more diverse and large scale datasets.

In evaluating these bounds, we assumed that the common factor $\frac{\|X\|_2}{\gamma\sqrt{m}}$ was the same and equal to 1 for a specific network across all bounds. Moreover, their growth rate is determined by assuming $\|W_d\|_2 = 1$ which factors out the common term related to the product of spectral norms. Furthermore, since we compare the bounds on models with the same architecture, and the bound in Theorem 3 is specifically derived to demonstrate a tighter depth dependency, we ignore the terms that depend solely on width or the rank of the matrix (i.e., the *pr* term in (19)). We evaluated the following bounds after training the networks on the different datasets to assess their tightness and overall performance:

- $\prod_{i=1}^{D} \|W_i\|_2 \sqrt{\sum_{i=1}^{D} \frac{\|W_i\|_F^2}{\|W_i\|_2^2}}$, Bound 1a (ours, General-Theorem 2)

- $\prod_{i=1}^{D} \|W_i\|_2 \sqrt{\sum_{i=1}^{D} \frac{\|W_i - W_{i_0}\|_F^2}{\|W_i\|_2^2}}$, Bound 1b (ours, Initialization based-Theorem 2)

- $\prod_{i=1}^{D} \|W_i\|_2 \sqrt{log\left(2Dpr + \sum_{i=1}^{D} \frac{\|W_i\|_F}{\|W_i\|_2 \sqrt{2}}\right)}$, Bound 1c (ours, specialized-Theorem 4)

- $\prod_{i=1}^{D} \|W_i\|_2 * \sqrt{k \sum_{i=1}^{D} n_i}$, Bound 2 (Tighter bound proposed in (Li et al., 2019))

- $\prod_{i=1}^{D} \|W_i\|_2 \sqrt{\left(\sum_{i=1}^{D} \frac{\|W_i\|_{2,1}^{2/3}}{\|W_i\|_2^{2/3}}\right)^3}$, Bound 3 (spectrally normalized margin bound of (Bartlett et al., 2017))

- $\prod_{i=1}^{D} \|W_i\|_2 \sqrt{D^2 p\left(\sum_{i=1}^{D} \frac{\|W_i\|_F^2}{\|W_i\|_2^2}\right)}$, Bound 4 (PAC-Bayesian bound proposed in (Neyshabur et al., 2018))

- $\prod_{i=1}^{D} \|W_i\|_F * min\{log(\frac{\prod_{i=1}^{D} \|W_i\|_F}{\Gamma}), \sqrt{D}\} \approx \prod_{i=1}^{D} \|W_i\|_F \sqrt{D}$, Bound 5 (size-independent bound of (Golowich et al., 2018))

In our first set of experiments conducted on the CIFAR-10 dataset using VGG and ResNets of different depths, we initialized the weights of the networks using Glorot-uniform initialization (Glorot & Bengio, 2010). During training, we constrained the filters to be of unit norm in all the models, following Li et al. (2019). We trained the network for 60 epochs with the SGD optimizer with 0.9 momentum and an initial learning rate of 0.01, which was decayed by half after the first 20 epochs, followed by the same decay of 0.5 every 10 epochs thereafter. The values of the different bounds for the different models for this set of experiments are compared in Table 2.

After an evaluation of the bounds for networks of different depths, we found that our bound in the general case (Bound 1a) was of order $10^7$ for the VGG-16 network, whereas bounds 3 and 4 were of order $10^8$, respectively, while Bound 5 (Golowich et al., 2018) was of order $10^{17}$ which was extremely loose due to its dependence on product of Frobenius norm across layers instead of the spectral norm. Furthermore, our bound based on the distance from initialization (Bound 1b) of the weights was even tighter than our general Bound 1a on the same network. This bound was tighter than even the recently proposed bound by Li et al. (2019) for the same network, despite the latter's more favorable size dependence. This demonstrates that the distance of weights from initialization can provide a tighter characterization of the generalization gap compared to regular norms. In addition, our sub-logarithmic bound (Bound 1c) was even tighter and was of the order $10^6$, which was smaller by a factor of almost $10^2$ compared to that of bounds 3 and 4.

Similarly, on a VGG-21 network, our general bound (Bound 1a) was of order $10^9$, while Bounds 3 and 4 were of order $10^{10}$, and Bound 5 was of order $10^{24}$. Moreover, on the same network, our Bound 1b was again

only 40% of that of Bound 1a, and our special bound (Bound 1c) was of the order of $10^7$ and was smaller than bounds 3 and 4 by an order of $10^3$ and Bound 2 by an order of 10. Similar trends were observed in the VGG-26 network where our bounds were consistently tighter and outperformed the other bounds, providing considerably tighter generalization estimates.

Thus, our general bound (Bound 1a) was consistently smaller by a factor of 10 compared to Bounds 3 and 4, on all variants of the VGG network. Moreover, our bound with the distance from initialization term (Bound 1b) was smaller than the value of our general bound (Bound 1a) and was about 100 times smaller than bounds 3 and 4 on most VGG network architectures. Most critically, our bounds were smaller than both, the size-independent results derived by Golowich et al. (2018) (Bound 5) and the recently obtained tighter bounds in Li et al. (2019) (Bound 2) by a factor of almost 10.

Similar trends were observed in the case of ResNet-18 and ResNet-34 models trained on the CIFAR-10 dataset with unit-norm constraint. On the ResNet-18 model, our initialization-based bound (1b) ($\sim 10^7$) was smaller by a factor of 100 from that of the bounds 3 and 4 ($\sim 10^9$) as can be observed from Table 2. Furthermore, our bound (1c) was even lower by a factor of 100 ($\sim 10^6$) compared to our general bound (1a), and by a factor of $10^3$ compared to bounds 3 and 4. Although the recently proposed bound 2 by (Li et al., 2019) is tighter than our general bound (1a), it is greater than our specialized bound with logarithmic depth dependence (1c) by a factor of 100. Similar observations were made in the case of Resnet-34 model as well. These results demonstrate the superiority and tightness of our bounds for more general architectures such as the ResNet.

In our second set of experiments, whose results are shown in Table 3, we evaluated and compared the different bounds on the models with different architectures on more diverse datasets, such as Tiny-imagenet and CIFAR-100, which contain hundreds of different categories. In this set of experiments, we do not apply any constraint on the norm of the networks as done the first set.

From the Table 3, we can again observe that our general bound (1a) was smaller than bounds 3,4, and 5, and the initialization-based variant of it (1b) was even smaller on all the datasets. Our specialized bound (1c) was again the tightest and was a factor of $10^3$ smaller than the bounds 3 and 4, and was significantly smaller than the bound 5. Although bound 2 was in some cases tighter than our general bound (1a), it was looser than our specialized bound (1c) by a factor of $10 - 100$. These observations demonstrate the wide applicability and generalizability of our bounds to more diverse architectures, general training settings without constraints, and real-world datasets.

The general variant of our bound (1a) is tighter than the bounds 3 and 4 because the term inside the square root depends solely on the sum of the ratio of the $L_{2,1}$ norm and the spectral norm of the weights in each layer, which grows only linearly as $O(D)$ with an increasing number of layers. On the other hand, in Bound 3, the same term, being a cube of the sum of the ratio of the two weight norms, grows at a faster rate of $O(D^3)$, which is proportional to the cube of the depth of the network. Similarly, the term in Bound 4 has a factor $D^2 p$ multiplied to this sum of norm-ratios term, and thus grows with a rate $O(D) \times D^2 p = O(D^3 p)$, which is much larger than the rate at which our bound grows.

The results of these experiments confirm that our bounds are tighter than the current state-of-art norm-based generalization bounds for deep networks. Furthermore, our bound (1c) is even tighter and has better dependence on network dimensions, growing sub-logarithmically with depth. Furthermore, our bound based on the distance of network weights from their initialization is always tighter in magnitude compared to the vanilla bound based on the Frobenius norm of the weights. This observation provides empirical evidence that the network weight travels only a small distance from initialization in overparameterized deep networks, and this traversed distance determines the generalization gap.

## 5    Discussion

In this section, we critically examine and compare the tightness of our proposed generalization bounds against the related bounds discussed above based on the observed empirical results. As shown in Tables 2 and 3, Bounds 3 and 4 were several orders of magnitude larger (10-100 times) than our proposed general bound (Bound 1a) for all the networks, and the difference was even more pronounced for our specialized

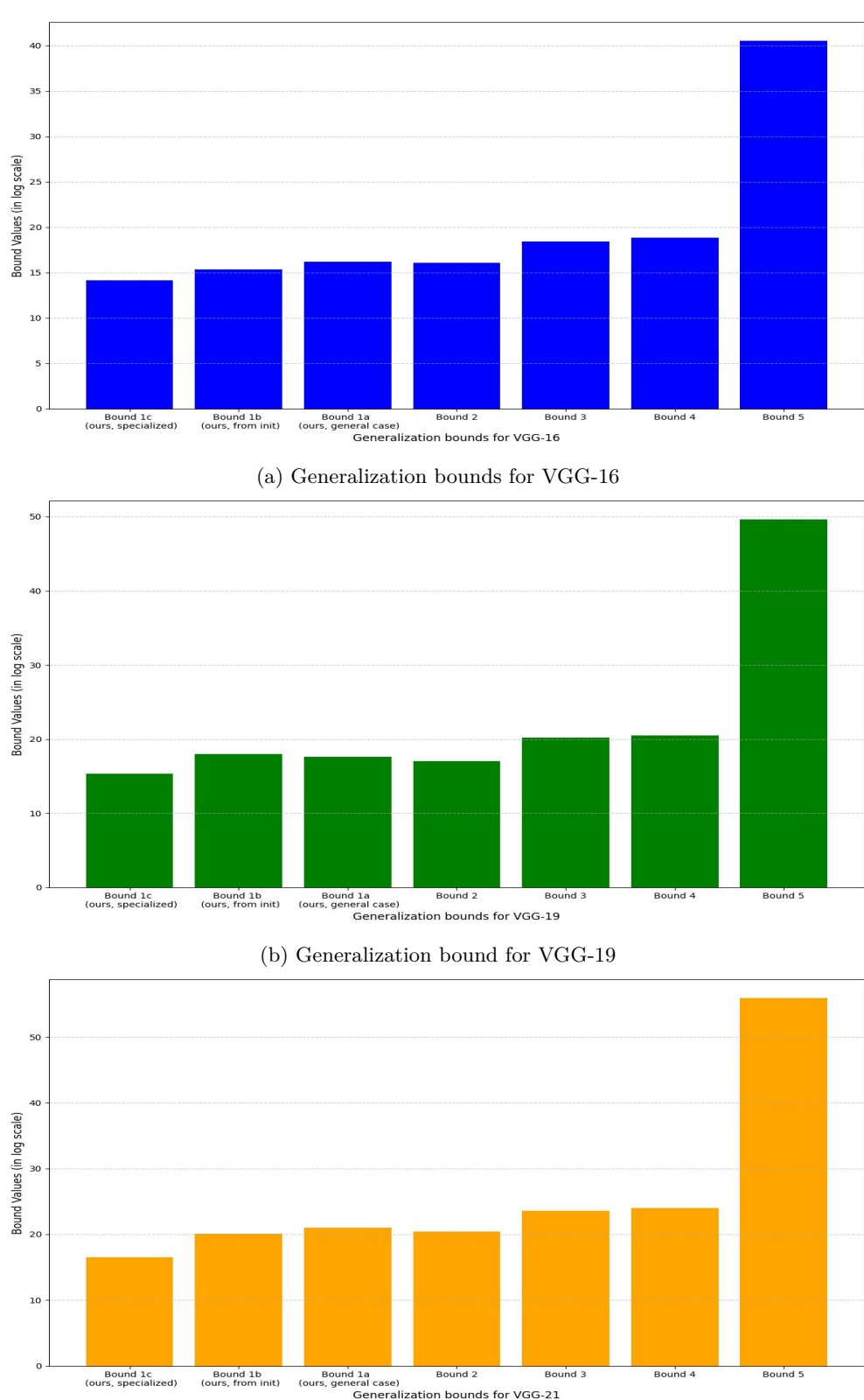

(a) Generalization bounds for VGG-16

(b) Generalization bound for VGG-19

(c) Generalization bound for VGG-21

Figure 1: Figure illustrating the values of the generalization bounds on a logarithmic scale for VGG-style networks of different depths (VGG-16 (subfigure a), VGG-19 (subfigure b), and VGG-21 (subfigure c)) (Bartlett et al., 2017) is much larger and loose for this network and is several orders larger than our bounds.

Table 2: Comparison of different generalization bounds with our nonvacuous bounds on VGG and ResNets with different depths and normalized weight matrices. Our specialized bound in the finite width case (Bound 1c) is the tightest of all bounds and is several order smaller than the previous bounds (Bounds 3 and 4).(Lecun et al., 1998)

| Model | Bound 1a | Bound 1b | Bound 1c | Bound 2 | Bound 3 | Bound 4 | Bound 5 |
|---|---|---|---|---|---|---|---|
| VGG-16 | $1.1 \times 10^7$ | $4.6 \times 10^6$ | $1.34 \times 10^6$ | $4.5 \times 10^7$ | $1 \times 10^8$ | $1.5 \times 10^8$ | $4.1 \times 10^{17}$ |
| VGG-19 | $4.5 \times 10^7$ | $6.3 \times 10^7$ | $4.5 \times 10^6$ | $2.44 \times 10^7$ | $5.8 \times 10^8$ | $8 \times 10^8$ | $3.68 \times 10^{21}$ |
| VGG-21 | $1.3 \times 10^9$ | $5.1 \times 10^8$ | $1.53 \times 10^7$ | $7.1 \times 10^8$ | $1.89 \times 10^{10}$ | $2.6 \times 10^{10}$ | $1.9 \times 10^{24}$ |
| VGG-26 | $2.9 \times 10^9$ | $1.6 \times 10^9$ | $7.1 \times 10^7$ | $1.1 \times 10^{10}$ | $9.4 \times 10^{11}$ | $3.6 \times 10^{11}$ | $9.17 \times 10^{30}$ |
| Resnet-18 | $1.2 \times 10^8$ | $5 \times 10^7$ | $4.99 \times 10^6$ | $2.3 \times 10^7$ | $1.4 \times 10^9$ | $2.5 \times 10^9$ | $3.0 \times 10^{23}$ |
| Resnet-34 | $1.1 \times 10^{10}$ | $3.6 \times 10^9$ | $3.2 \times 10^8$ | $4.67 \times 10^9$ | $2.5 \times 10^{11}$ | $4.2 \times 10^{11}$ | $3.3 \times 10^{41}$ |

Table 3: Comparison of different generalization bounds with our bounds on VGG and Residual networks without any norm constraint on diverse image datasets. Our bound (1c) is again the tightest of all bounds and is several order smaller than the previous bounds (Bounds 2, 3 and 4).(Lecun et al., 1998)

| Dataset | Model | Bound 1a | Bound 1b | Bound 1c | Bound 2 | Bound 3 | Bound 4 | Bound 5 |
|---|---|---|---|---|---|---|---|---|
| CIFAR-10 | VGG-19 | $9.2 \times 10^{22}$ | $8.3 \times 10^{22}$ | $3.3 \times 10^{21}$ | $4.0 \times 10^{21}$ | $1.11 \times 10^{24}$ | $1.69 \times 10^{24}$ | $9.66 \times 10^{30}$ |
| CIFAR-100 | ResNet-34 | $1.60 \times 10^{39}$ | $1.45 \times 10^{39}$ | $4.36 \times 10^{37}$ | $1.55 \times 10^{39}$ | $3.19 \times 10^{40}$ | $4.8 \times 10^{40}$ | $2.7 \times 10^{58}$ |
| Tiny Imagenet | ResNet-18 | $1.24 \times 10^{13}$ | $1.82 \times 10^{12}$ | $5.4 \times 10^{11}$ | $1.8 \times 10^{13}$ | $2.15 \times 10^{14}$ | $3.6 \times 10^{14}$ | $1.07 \times 10^{25}$ |

bound (Bound 1c), which was almost $10^3 \sim 10^4$ times smaller than these bounds. These observations align well with our theoretical predictions, according to which these bounds scale at a larger rate of $O(\sqrt{D^3 pr})$ than our Bounds 1a and 1c that scale at rates of $(O(\sqrt{Dpr}))$ and $(O(\sqrt{log\ Dpr}))$ respectively assuming that the spectral norm of weights in each layer is bounded by 1.

Surprisingly, our bounds demonstrated considerably tighter estimates, even when compared to the size-independent bounds (Bound 5) proposed by Golowich et al. (2018). We conjecture that this was due to the explicit dependence of this bound on the product of the Frobenius weight norm across all layers, which can grow exponentially faster than the product of spectral norm used in the other bounds (including ours). This is because the Frobenius norm is always larger than the spectral norm, since the Frobenius norm is proportional to the sum of the squares of all the singular values of the matrix, while the spectral norm is only dependent on the largest singular value. Moreover, for a layer $d$ with $p$ hidden units, the ratio of the Frobenius norm to the spectral norm satisfies the following:

$$\frac{\|W_d\|_F}{\|W_d\|_2} = \sqrt{\frac{\sum_i \sigma_i^2}{\sigma_1^2}} \le \sqrt{\frac{r\sigma_1^2}{\sigma_1^2}} = \sqrt{r} \le \sqrt{p}$$

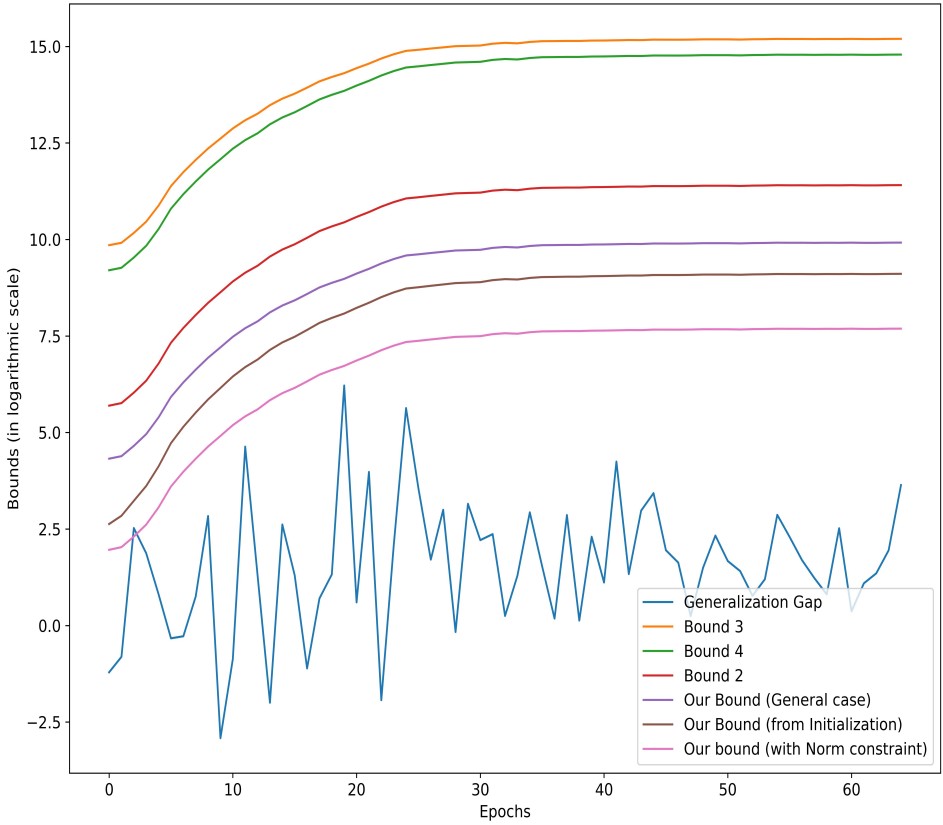

Figure 2: Figure illustrating the values of the different generalization bounds on a logarithmic scale as well as the generalization gap (difference between test and train loss) across different epochs for a VGG-21 model. Our proposed bound with the norm constraint on hidden units is the tightest bound and has the lowest value compared to the other bounds across all the epochs. On the other hand, Bound 3 (Bartlett et al., 2017) is much larger and loose for this network and is several orders larger than our bounds.

where $r$ denotes the rank of the weight matrix which is always less than $p$. Thus, for a network with $D$ layers, the ratio of the product of the two norms can grow exponentially according to the following inequality:

$$\frac{\prod_{d=1}^{D} \|W_d\|_F}{\prod_{d=1}^{D} \|W_d\|_2} \leq (\sqrt{p})^D \tag{20}$$

This fundamental difference in norm choice explains the faster growth rate of this bound with depth. Thus, this analysis leads us to posit that this bound's size independence relies upon the strong and often unrealistic assumption that the Frobenius norm of each layer be a very small quantity below 1, otherwise it risks an exponential increase with depth at a rate $r^D$, where $r>1$. This assumption is too restrictive and often may not hold in practical scenarios. In contrast, the tighter version of our bound (1c) is derived under a more relaxed and empirically plausible assumption: that the maximum width of any layer is finite and bounded, which holds well in practical scenarios. Thus, without applying any strong constraints on the norm of neural networks, we derive tighter bounds with sub-logarithmic depth dependence for overparameterized networks

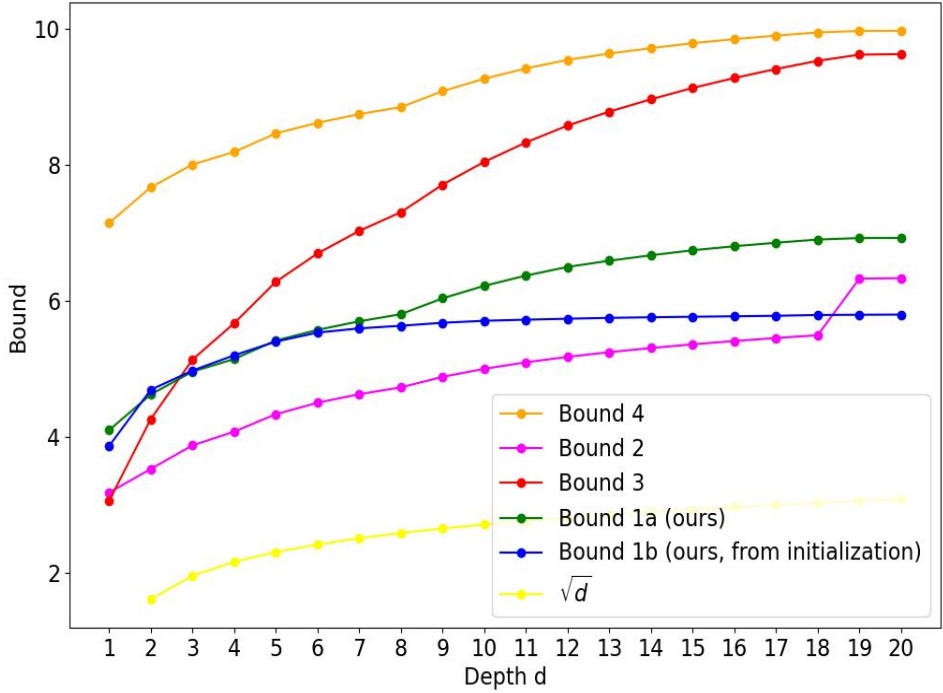

Figure 3: Figure illustrating the growth of different bounds with depth of a VGG-21 network. The horizontal axis in this figure corresponds to the depth, and the vertical axis represent the corresponding bound values in logarithmic scale.

than the more stringent constraint that is placed on the Frobenius norm of each layer placed in Bound 5 ((Golowich et al., 2018)) which may not hold in practice.

Furthermore, we observed that incorporating the distance of the weights from their initialization into the bounds always leads to tighter bounds compared to those that simply rely on the weight norm. This can be confirmed from Table 2, where we observe that the Bound 1b, which includes the distance from initialization term, was always smaller and less than half of the bound value in the general case (Bound 1a) for several networks of different depths.

More critically, our empirical analysis reveals that our bounds were much tighter than even the recently proposed bounds by Li et al. (2019) (Bound 2) that used a term that explicitly depends on the network dimensions ($\sqrt{Dpr}$), whereas our bounds are mostly dependent on the network norms. As observed in Table 2, our Bound 1b and 1c were smaller than this bound in most cases, verifying their tighter estimates empirically. This demonstrates that our bounds involving distance from initialization term (Bound 1b) and our specialized bound (1c) increase at a rate that is much slower than the square root of the number of network parameters. Furthermore, our specialized bounds for the norm-constrained case (Bound 1c) are almost 10-100 times smaller than Bound 2 for networks of all sizes, highlighting the much tighter estimates provided by our bounds.

The results of these experiments suggest that although the works in Golowich et al. (2018) and Li et al. (2019) (Bounds 5 and 2) attempted to provide bounds that were almost size-independent or scaled more favorably with network dimension, our experimental observations suggest that their bounds are either still loose or are predicated on the assumptions of a constrained Frobenius norm in each layer, which are often larger compared to the spectral norm or the $L_{2,1}$ norm used in our bounds. On the other hand, our bounds

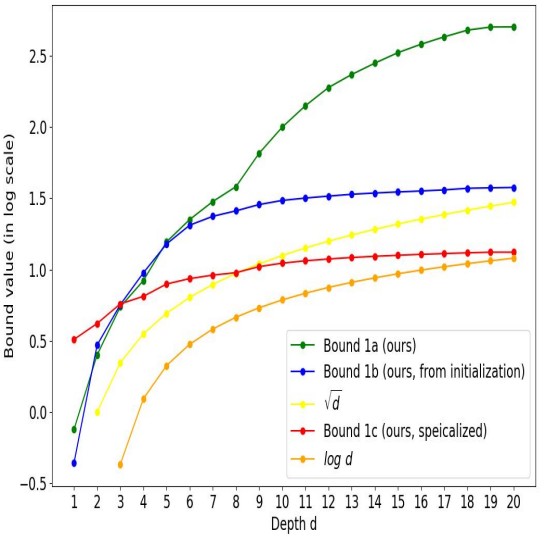 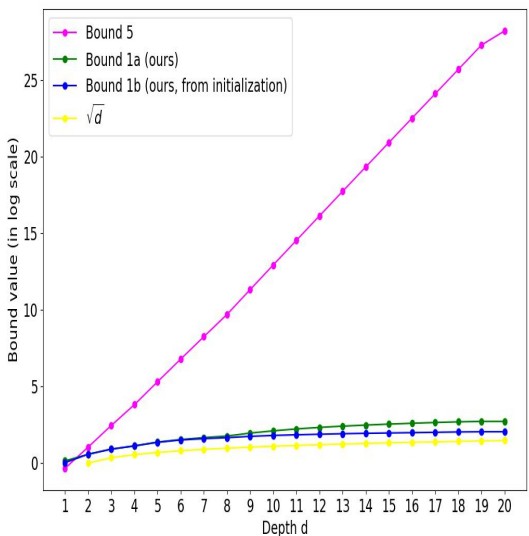

(a) Comparison of growth rate with depth of our specialized bound
(1c) with our other bounds (1a, 1b), the square root of depth, and the log of the depth for a VGG-21 network.

(b) Demonstration of the explosive growth rate of the Bound 5 with depth when compared to our bounds.

Figure 4: In the subfigure (a), we demonstrate the tight scaling behavior of our norm-constrained bound (1c) with respect to our other bounds across different layers of depths $d$ of a VGG-21 network. From the figure, we can see that our bound 1c grows at a smaller rate than $O(log\ d)$ and much slower than our other bounds as well as $O(\sqrt{d})$. In subfigure (b), we can observe that the Bound 5 (Golowich et al., 2018), which was purported to be size independent, grew at a much faster rate than our other bounds demonstrating its poor scaling.

not only provide more favorable scaling, but also provide much tighter estimates in the bounded norm case. Furthermore, when deriving our logarithmic bounds, we do not impose any bounds on the weight norms of the weights in individual layers, which is significantly less restrictive and more realistic than the stringent constraint on the norm of the entire layer imposed in (Golowich et al., 2018) to derive their bounds.

## 6 Conclusion

In this work, novel nonvacuous generalization bounds demonstrating better dependence on network dimensions (width as well as depth) were introduced.

To provide a motivation for our bounds, we began by outlining the shortcomings of traditional complexity bounds, such as those based on VC dimension and the principle of uniform convergence, which often yield vacuous bounds for deep over-parameterized networks. We then proceeded to provide an overview of prior nonvacuous bounds that attempted to overcome these shortcomings. These bounds were functions of the network norms and their margin. We discussed their strengths while also underscoring a significant limitation: their poor scaling with network dimensions either in the form $O(\frac{\sqrt{D^3 pr}}{\sqrt{m}})$ or $O(r^D)$. We then introduced our bounds that provided significantly tighter estimates with better dependence on network dimensions (scaling with a rate of $O(\sqrt{Dpr})$). Moreover, under specific justifiable assumptions regarding the network width, our bounds achieved a remarkable sub-logarithmic scaling of $(O(\frac{\sqrt{log\ D}}{\sqrt{m}})$ with depth for a fixed maximum width $p$ and maximum rank $r$.

The theoretical rigor of our work is supported by extensive mathematical proofs of our bounds for both the general and the special cases. Empirical validation on VGG and ResNets of different depths on diverse datasets demonstrated the significantly tighter estimates and better depth scaling provided by our bounds compared to the competitive nonvacuous bounds.

Our rigorous theoretical framework, combined with compelling experimental evidence, established the superiority of our bounds compared to the previous ones and also demonstrates the tighter depth as well as width dependence of our bounds compared to the previous work. Thus, our work provides novel theoretical insights into the remarkable generalization capabilities of very deep networks, especially in the over-parameterized regime. Our findings contribute to a deeper understanding of the reason behind the superior generalization performance of deep neural networks, despite their immense complexity.

## 7   Future directions

Our current work provides a robust foundation for a deeper understanding of generalization in deep networks, yet it also opens several avenues for future research to derive even tighter bounds that may provide a more precise estimate of network generalization.

One critical direction is the incorporation of the role of implicit regularization provided by Stochastic Gradient Descent (SGD). Although our derived bounds exclude the effect of this crucial component, many works have demonstrated that SGD indeed plays a vital role in biasing the trained network towards lower complexity solutions (Smith et al., 2021; Arora et al., 2019). It would be an interesting direction to analyse the effects of this optimization algorithm in greater detail and to include its impact on the generalization bound for the deep networks.

Furthermore, as shown in (Zhang et al., 2021) data plays an important role in determining how well the trained network will generalize to unseen data. As (Zhang et al., 2021) has shown that deep networks have sufficient capacity to fit even random data, but a network trained on such data will obviously generalize very poorly, the data distribution thus plays a significant role in determining the generalization property of the deep networks. It would be thus an important direction to explore the impact of the data distribution on the generalization bounds of the neural networks. Thus, in future works, we would like to extend our bounds to address these challenges and incorporate the role of the training algorithm and the data distribution in the derivation of more tighter and improved generalization bounds.

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

# A   Appendix

## A.1   Additional Lemmas

In this section of the appendix, we will derive some additional lemmas which will be useful in deriving our generalization bound in the special norm-constrained case.

**Lemma A1.** *Assume $(\epsilon_1, \epsilon_2, ..., \epsilon_D)$ are given along with a set of matrices $W_1, W_2, .., W_i, .., W_D$ for a D layer neural network that lie within $\mathcal{B}_1 \times \mathcal{B}_2 \times .. \times \mathcal{B}_D$ where $\mathcal{B}_i$ are classes in $\mathbb{R}^p$ with variable p. and $\|W_i\|_2 < c_i$. Then the hypothesis class representing the neural network outputs ($\mathcal{F}_\mathcal{D} = \{Z_\mathcal{W} : \mathcal{W} \in \mathcal{B}_1 \times \mathcal{B}_2 \times \mathcal{B}_3 \times .. \times \mathcal{B}_D\}$) has the covering bound.*

$$\mathcal{N}(\mathcal{F}_\mathcal{D}, \delta, |.|_L) = \prod_{i=1}^{D} \sup_{W_1, W_2, .., W_{i-1}} \mathcal{N}\Big(\{W_i Z_{i, W_1, W_2.., W_{i-1}}(.) : W_i \in B_i\}, \epsilon_i, \|.\|\Big) \tag{21}$$

**Proof.** To prove this theorem, we will use the bound derived in (Bartlett et al., 2017) to achieve a tighter bound. We will break down the task of calculating the covering number for the $i_{th}$ layer into a product of the covering number for the previous layer $(i-1)$ and the cover for the weights of the $i_{th}$ layer. We will proceed with the proof inductively.

- Firstly, let's consider the covering bound for the first layer defined by the class $\mathcal{F}_1 = \{W_1 X : W_1 \in \mathcal{B}_\infty\}$ and denote it by $N_1 = \mathcal{N}(\mathcal{F}_1, \epsilon_1, |.|_2)$

- Assuming that the lemma holds for all layers up to $i$, for each function $F$ in the function class $\mathcal{F}_i$ in layer $i$, define $\mathcal{H}_{i+1}(F)$ with an $\epsilon_{i+1}$ cover of $\{W_{i+1} Z_{W_1, W_2.., W_{i-1}, W_i} : W_{i+1} \in \mathcal{B}_{i+1}\}$, construct a cover for the function of $i+1$ the layer individually defined as

$$|\mathcal{H}_{i+1}(F)| \leq \sup_{(W_1, W_2, W_3, .., W_i)} \mathcal{N}(\{W_{i+1} Z_{W_1, W_2, .., W_i} : W_{i+1} \in \mathcal{B}_{i+1}\}, \epsilon_{i+1}, \|.\|_p) := N_{i+1} \tag{22}$$

  Now, using above, taking a union bound over function in $\mathcal{F}_i$, we can define the cover for the $(i+1)_{th}$ layer as follows

$$\mathcal{F}_{i+1} = \bigcup_{F \in \mathcal{F}_i} \mathcal{H}_{i+1} \tag{23}$$

$$|\mathcal{F}_{i+1}| \leq |\mathcal{F}_i| * |\mathcal{H}_{i+1}| \leq |\mathcal{F}_i| * N_{i+1} \leq N_{i+1} N_i |\mathcal{F}_{i-1}| \leq \prod_{j=1}^{i+1} N_j \tag{24}$$

Thus, this shows that the covering number of the whole network can be decomposed into the product of the covering number of the different layers.

**Lemma A2.** *Given a neural network with $D$ layers with a maximum of $p$ hidden units in any layer. The covering number of the $i_{th}$ layer with a rank $r$ of its weight matrix is given by*

$$\left(1 + \frac{2b_i L_{W_i}}{\epsilon_i}\right)^{2pr} \tag{25}$$

*where $L_{w_i}$ is the Lipschitz constant of the $i_{th}$ layer, $b_i$ bounds the Frobenuius norm of weights of the $i_{th}$ layer $\|W_i\|_F$, and $\epsilon_i$ denotes the radius of the cover element used in the $i_{th}$ layer.*

**Proof of the lemma**

Let us assume that the output of the $i_{th}$ layer of the neural network with at most $p$ hidden units is given by $f^i(w, x)$, where $f^i(.)$ belongs to the function space $f_i \in \mathcal{F}$. Then, the metric in this function space is given as

$$L_g = \sup_x \|f_1^i(x) - f_2^i(x)\|_2 \le \sup_x \|f(w_1^i, x) - f(w_2^i, x)\|_2 \le L_{w_i} \|w_1^i - w_2^i\|_2 \tag{26}$$

with Lipschitz constant $L_{w_i}$ and the weights in each layer are bounded as $\|W^i\|_{2,1} < b_i$. We can now bound the difference of function output in (26) for row-wise changes in weight vectors and sum it across all rows to get the following inequality:

$$L_g = \sup_x \|f_1^i(x) - f_2^i(x)\|_2 \le \sup_x \|\sigma(W_1^i x) - \sigma(W_2^i x)\|_2 \le \sup_x \left\| \begin{array}{c} \sigma'(W_{21}^i x)(W_{11}^i x - W_{21}^i x) \\ \sigma'(W_{22}^i x)(W_{12}^i x - W_{22}^i x) \\ \vdots \\ \sigma'(W_{2p}^i x)(W_{1p}^i x - W_{2p}^i x) \end{array} \right\|_2 \tag{27}$$

Assuming that the activation function $\sigma$ has a Lipschitz constant $\rho$, then the derivative $\sigma'$ is upper bounded by $\rho$ and the above can be simplified as follows:

$$\sup_x \left\| \begin{array}{c} \rho(W_{11}^i x - W_{21}^i x) \\ \rho(W_{12}^i x - W_{22}^i x) \\ \vdots \\ \rho(W_{1p}^i x - W_{2p}^i x) \end{array} \right\|_2 \le \sup_x \|\rho(W_1^i - W_2^i)x\|_2 \overset{(i)}{\le} \sup_x \rho\|W_1^i - W_2^i\|_2 \|x\|_2$$

$$\overset{(ii)}{\le} \rho\|W_{1j}^i - W_{2j}^i\|_F * \sup_x \|x\|_2 \le \rho\|W_1^i - W_2^i\|_F R \le \rho R\|W_{1j}^i - W_{2j}^i\|_F \tag{28}$$

where inequality $(i)$ in the above expression is obtained by using the inequality $\|AB\|_2 \le \|A\|_2 \|B\|_2$, inequality $(ii)$ is obtained using that fact that the Frobenious norm is an upper bound on the spectral norm $\|A\|_2 \le \|A\|_F$, and $R = \sup_x \|x\|_2$. Assuming that the layer-wise Lipschitz constant is defined as $L_{w_i} = \rho R$, we get:

$$L_g \le \rho R\|W_1^i - W_2^i\|_F \le \rho R\|W_1^i + (-W_2^i)\|_F \le \rho R(\|W_1^i\|_F + \|W_2^i\|_F) \le L_{w_i} * 2b_i \tag{29}$$

Now, since we know the maximum value that can be achieved by the layer-wise metric $L_g$, the problem of finding the covering number for the function space can be transferred to finding that of finding the covering number for the weight space $\mathcal{W}$ that satisfies the bound $\|W^i\|_F \le b_i$.

From (Vershynin, 2018), we know that the $\epsilon$-covering number of a set $A \in \mathbb{R}^p$ with diameter $D_A$ is bounded by $\left(1 + \frac{2r_A}{\epsilon}\right)^p = \left(1 + \frac{D_A}{\epsilon}\right)^p$.

To understand this intuitively, imagine a sphere with a diameter of side $D_A$ in $p$ dimensions, hence its volume will be $\frac{\pi^{p/2}}{\Gamma(p/2+1)}(D_A/2)^p$. Considering balls of radius $\epsilon$ have a volume $\frac{\pi^{p/2}}{\Gamma(p/2+1)}\epsilon^p$, the total number of such balls required to cover this larger sphere would be $\frac{(D_A)^p}{\epsilon^p} = (\frac{D_A}{2\epsilon})^p = (\frac{r}{2\epsilon})^p$. However this is not an upper bound, because the radius of the larger sphere might not be an integer multiple of the smaller

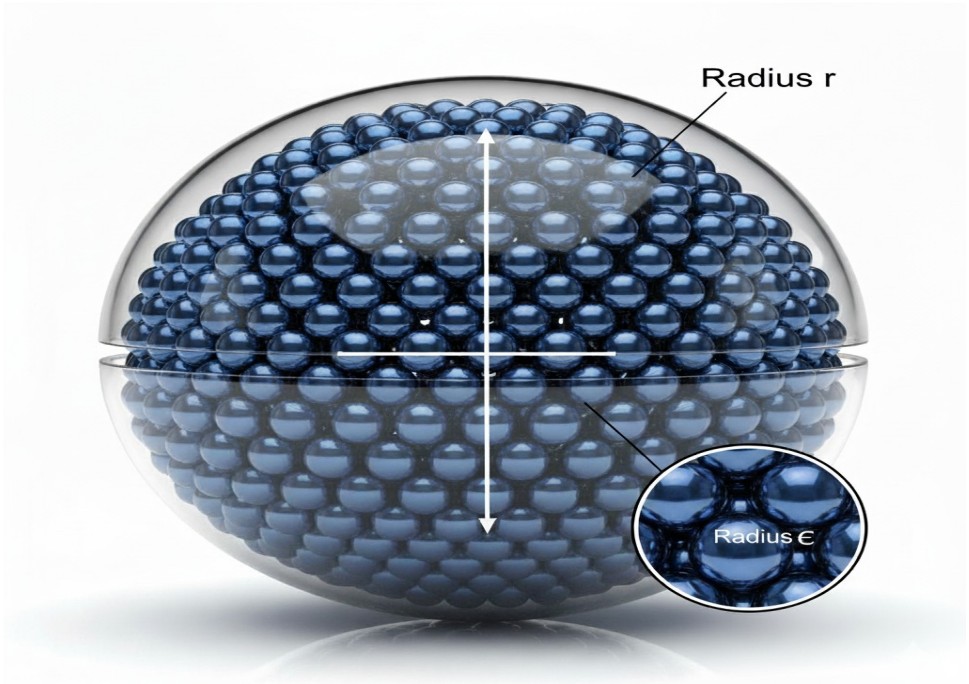

Figure 5: Figure illustrating the concept of covering number in a three dimensional functional space. Considering the larger sphere to be the space of all possible functions, we want to pack it with small spheres such that every function is at most at a distance of $\epsilon$ from one of those spheres. Since the volume of the large sphere is $\frac{4}{3}\pi r^3$ and the volume of smaller sphere $= 4/3\pi\epsilon^3$, the number of such spheres required would be $(\frac{r}{\epsilon})^3$

one and the value might become fractional which is not possible. Hence, to overcome this problem, we extend the radius of larger sphere to $r + \epsilon/2$ and pack it with non-overlapping spheres of size $\epsilon/2$. Defining a sphere with radius $x$ as $B(x)$, the number of small spheres required to cover this extended sphere is $vol(B(r + \epsilon/2))/vol(B(r + \epsilon/2)) = (\frac{r+\epsilon/2}{\epsilon/2})^d = (1 + \frac{2r}{\epsilon})^d$, which is the required upper bound.

For our weight space $\mathcal{W} \in \mathbb{R}^{p\times d}$ with a rank of $r$, the diameter is bounded as $D_A = \sup_{W_1^i, W_2^i} \|W_1^i - W_2^i\|_F =$

$\sup_{W_2^i, W_2^i} \|W_1^i + (-W_2^i)\|_F \overset{(i)}{\leq} \sup_{W_1^i, W_2^i} \|W_1^i\|_F + \|W_2^i\|_F \leq 2\sup_{W_1^i}\|W_1^i\|_F = 2 * b_i$, where $(i)$ is obtained by applying the triangle inequality. Hence, the upper bound on the $\epsilon$-covering number over the weight space $\mathcal{W}$ in layer $i$ is given by $\left(1 + \frac{D_A}{\epsilon}\right)^{pd} = \left(1 + \frac{2b_i}{\epsilon}\right)^{pd}$ or $\left(1 + \frac{2b_i}{\epsilon}\right)^{D_{eff}}$ where $D_{eff}$ is the effective dimensionality.

For a weight matrix with rank $r$, input dimension $d$ and $p$ hidden units, the effective dimensionality is $D_{eff}$ $= r(p + d - r)$. This is because any such matrix can be decomposed as $W = UV^T$, where $U$ is a matrix of dimension $p \times r$ and $V$ is a matrix of dimension $d \times r$, and both are of full rank with total effective

dimensionality $= pr + dr$. Moreover, this decomposition is not unique, since we can multiply an invertible matrix $A$ of rank $r \times r$ to obtain $W = (UA^{-1})(AV^T)$, all of which lead to the same decomposition. Thus, we will need to subtract $r^2$ from the effective dimensionality. Hence, the net effective dimensionality is $D_{eff} = pr + dr - r^2$.

Incorporating $D_{eff} = r(p + d - r)$, this bound can be written as $\left(1 + \frac{2b_i}{\epsilon_i}\right)^{r(p+d-r)} \overset{(iii)}{\leq} \left(1 + \frac{2b_i}{\epsilon_i}\right)^{2pr-r^2}$. (Here, inequality $(iii)$ is derived using the facts that the network has a uniform width and the maximum no of hidden units $= d \leq p$.

Using the result for $\epsilon-$covering on the weight space, we can now translate this result to the function space $f \in \mathcal{F}$ as follows. Let us consider a set $C = \{W_1, W_2, ..W_N\}$ to be an $\epsilon - cover$ of the weight space. For each function $f^i(W, x)$ in the function space, we can consider a center function $f^i(W_c, x)$ with weights $W_c$ belonging to the cover $C$ such that $L_f(f(W;x), f(W_c;x)) \leq L_{w_i}\|W - W_c\|_{2,1} \leq L_{w_i}\epsilon$, which is the smallest distance of a function to the nearest central function. Now, to make a $\delta$-cover in the function space, we need $L_{w_i}\epsilon \leq \delta$, or $\epsilon \leq \delta/L_{W_i}$. Thus, to get a $\delta_i$ cover in the function space, we will need a corresponding cover in weight space with $\epsilon = \delta_i/L_{W_i}$. Substituting this value of $\epsilon$ in our covering bound for the weight space, we get:

$$\mathcal{N}(\mathcal{F}, \delta_i, L_g) = \mathcal{N}\left(\mathcal{W}, \frac{\delta_i}{L_{w_i}}, \|.\|_{2,1}\right) \leq \left(1 + \frac{2b_i}{\delta_i/L_{w_i}}\right)^{2pr-r^2} = \left(1 + \frac{2L_{w_i}b_i}{\delta_i}\right)^{2pr-r^2} \overset{(i)}{\leq} \left(1 + \frac{2L_{w_i}b_i}{\delta_i}\right)^{2pr} \quad (30)$$

Here, inequality $(i)$ holds because the term inside the bracket is larger than 1 ($b_i$ being a norm, $\delta_i$ being a radius, and $L_{w_i}$ being the Lipschitz constant are all greater than 1). Replacing the variable $\delta_i$ with $\epsilon_i$ in the above inequality, we finally complete the proof of Lemma A2.

Furthermore, we will use the following lemma to bound the Lipschitz constant $(L_{w_i})$ of the $i_{th}$ layer of the network:

**Lemma A3.** *Given a neural network $f_W(.)$ whose Jacobian up to the $d_{th}$ layer is uniformly upper bounded by $B_{1:d}^{jac}$ and the maximum input norm is bounded by $\max_x \|x\|_2 < R$ then the Lipschitz constant of the $d_{th}$ layer output with respect to the $d_{th}$ layer weights $w_d$ can be upper-bounded as*

$$L_{w_i} < \rho_i B_{1:d-1}^{jac} R \quad (31)$$

**Proof of the Lemma**

Given the function representing the neural network with $L$ layers as $f_{W_L}(f_{W_{L-1}}(..f_{W_d}..(f_{W_1}(x))..)..)$. We can derive the Lipschitz constant of the output of $d_{th}$ layer with respect to its weights by considering the change in the loss function for the different configurations of the weights in layer $d$:

$$\sup_{W_{d1}, W_{d2}} \|f_{W_{d1}}(..(f_{W_1}(x))..) - f_{W_{d2}}(..(f_{W_1}(x))..)\|_2$$

$$\overset{(i)}{\leq} \sup_{W_{d1}, W_{d2}} \|f_{w_{d1}}(\rho_{d-1}\|W_{d-1}\|_2\rho_{d-2}\|W_{d-2}\|_2...\rho_1\|W_1\|X)$$
$$- f_{W_{d2}}(\rho_{d-1}\|W_{d-1}\|_2\rho_{d-2}\|W_{d-2}\|_2...\rho_1\|W_1\|X)\|_2$$

$$\overset{(ii)}{\leq} \sup_{W_{d1}, W_{d2}} \|\rho_d W_{d1}(\rho_{d-1}\|W_{d-1}\|_2\rho_{d-2}\|W_{d-2}\|_2...\rho_1\|W_1\|) - \rho_d W_{d2}(\rho_{d-1}\|W_{d-1}\|_2\rho_{d-2}\|W_{d-2}\|_2...\rho_1\|W_1\|)\|_2\|X\|_2$$

$$\overset{(iii)}{\leq} \sup_{W_{d1}, W_{d2}} (\rho_d W_{d1}\|J_{d-1}J_{d-2}...J_1\| - \rho_d W_{d2}\|J_{d-1}J_{d-2}...J_1\|_2)\|\|X\|_2$$

$$\leq \sup_{W_{d1}, W_{d2}} (\rho_d W_{d1} - \rho_d W_{d2})\|J_{d-1}J_{d-2}...J_1\|_2\|X\|_2 \leq (\rho_d(W_{d1} - W_{d2}))\|J_{d-1}J_{d-2}...J_1\|_2\|X\|_2$$

$$\leq \sup_{W_{d1}, W_{d2}} \rho_d B_{1:d-1}^{jac}\|W_{d1} - W_{d2}\|_2\|X\|_2 = L_{W_d}\|W_{d1} - W_{d2}\|_2 \quad (32)$$

Here, $(i)$ is obtained by decomposing the function of the $d-1_{th}$ layer into the Lipschitz constant of the prior layers $f_{w_{d-1}} = \sigma_{d-1}(W_{d-1}\sigma_{d-2}(..)..\sigma_1(W_1 x + b)) = \rho_{d-1}\|W_{d-1}\|_2\rho_{d-2}\|W_{d-2}\|_2..\rho_1\|W_1\|_2$ The inequality $(ii)$ is derived from $(i)$ by factoring out the common term $\|X\|_2$ and using the fact that $\|f_{w_{d1}}(x) - f_{w_{d2}}(x)\|_2 \leq \|\rho_d(W_{d1}\prod_{i=d-1}^1 \rho_i\|W_{i-1}\|_2 - W_{d2}\prod_{i=d-1}^1 \rho_i\|W_{i-1}\|_2)X\|_2 \leq \rho_d\|W_{d1} - W_{d2}\|_2\|X\|_2 \prod_{i=d-1}^1 \rho_i\|W_{i-1}\|_2$, where $\rho_{d+1}$ is the Lipschitz constant for the $d_{th}$ layer. Inequality $(iii)$ is obtained by replacing the Lipscitz constant of each layer $\rho_d\|W_d\|_2$ with the layer-wise Jacobian $J_d$. Comparing the value of $L_{w_d}$ in the the last step of the above derivation with its left hand side, we get the Lipschitz constant of the $d_{th}$ layer weights as follows

$$L_{w_d} = \rho_d B_{1:d-1}^{jac} R \tag{33}$$

where, $B_{1:d-1}^{jac}$ is the uniform upper bound of the Jacobian of the output of $d_{th}$ layer with respect to its weights, and we substitute $\max_x \|X\|_2 = R$. This completes the proof of our lemma.

## A.2 Further important proofs

We now provide the proofs of the important lemmas and theorems stated earlier in the paper.

### A.2.1 Proof of our initialization-based PAC-Bayesian bound (Theorem 2)

To prove this theorem, we will first evaluate an upper bound on the maximum allowable perturbation $U_i$ on each layer such that it does not change the empirical risk and the function output by a specified threshold proportional to $\gamma$. Using this threshold, we will calculate the $KL(\mathbf{w} + \mathbf{u}\|\mathbf{P})$ term, which we substitute in Lemma 1 to obtain the final Theorem.

Let us assume $\beta = (\prod_{i=1}^D \|W\|_i)^{1/d}$. Since the ReLU activation function is homogenous, we can consider the normalized weights $\tilde{\mathbf{w}} = \beta W_i/\|W_i\|_2$ and the resulting output of the network $f_{\tilde{\mathbf{w}}} = f_{\mathbf{w}}$. Moreover, the expected and empirical loss will be the same in both cases. It can also be verified that the product of spectral norms remains same for both the weight, i.e., $\prod_{i=1}^D \|W_i\|_2 = \prod_{i=1}^D \|\widetilde{W_i}\|_2$ and that $\frac{\|W_i\|_F}{\|W_i\|_2} = \frac{\|\widetilde{W_i}\|_F}{\|\widetilde{W_i}\|_2}$. Thus, it is sufficient to prove this theorem for normalized weights for which the spectral norm is uniform across the layers and is equal to $\|W_i\|_2 = \beta$ and the result will then hold in the general case.

Now, let us consider the prior distribution on weights $P$ to be $\mathcal{N}(\mathbf{W_0}, \sigma^2 I)$ to derive the equation (13) from Theorem 2, where $\mathbf{W_0} = [W_{1_0}, W_{2_0}, .., W_{i_0}, .., W_{D_0}]$ denotes the weights at initialization. Then from Lemma 2 in (Neyshabur et al., 2018) since the perturbations in each layer follows a normal distribution and satisfy $\|U_i\|_2 \leq 1/d\|W_i\|_2$, the variance of $U_i$ should be $1/d$ times that of $W_i$. Hence, we can specify the distribution of $U_i$ as $\mathcal{N}(0, \frac{\sigma^2 I}{d^2})$, so that the distribution of each layer is $W_i + U_i \sim \mathcal{N}(W_i, \frac{\sigma^2 I}{d^2})$ and in general $\mathbf{w} + \mathbf{u} \sim \mathcal{N}(\mathbf{w}, \frac{\sigma^2 I}{d^2})$. Since the prior cannot depend on the learned weights $W$, we will assign $\sigma$ to be proportional to a term $\widetilde{\beta}$. We will select this $\widetilde{\beta}$ from a predetermined grid such that for each weight configuration, we will have a $\widetilde{\beta}$ satisfying $|\beta - \widetilde{\beta}| \leq \frac{1}{d}\beta$. This choice allows us to derive PAC-Bayesian bounds for different values of $\beta$, such that each $\beta$ is close to one of the $\widetilde{\beta}$. Subsequently, taking a union bound over the possible values of $\widetilde{\beta}$ will allow us to derive our specified bound.

Since $\mathbf{u} \sim \mathcal{N}(0, \frac{\sigma^2}{d}I)$, using Tropp's inequality (Tropp, 2011) on the matrix $U_i$, we obtain the following bound on its spectral norm.

$$\mathbb{P}_{U_i \sim \mathcal{N}(0, \sigma^2/D^2)}[\|U_i\|_2 \geq t] < 2pe^{\frac{-t^2}{p\sigma^2/D^2}} = 2pe^{\frac{-(Dt)^2}{p\sigma^2}} \tag{34}$$

Taking a union bound over the layers, we obtain

$$\mathbb{P}_{\forall_i U_i \sim \mathcal{N}(0, \sigma^2/D^2)}[\max_i \|U_i\|_2 \geq t] < 2pDe^{\frac{-t^2}{p\sigma^2/D^2}} = 2pDe^{\frac{-(Dt)^2}{p\sigma^2}} \tag{35}$$

Now, for this inequality to be satisfied with probability $\geq \frac{1}{2}$,

$$2pDe^{\frac{-(Dt)^2}{p\sigma^2}} < 1/2 \tag{36}$$

$$\implies ln(2pDe^{\frac{-(Dt)^2}{p\sigma^2}}) < ln(1/2) \tag{37}$$

$$\implies ln(2pD) - \frac{(Dt)^2}{p\sigma^2} < ln(1/2) \tag{38}$$

$$\implies ln(4pD) < \frac{D^2t^2}{p\sigma^2} \tag{39}$$

$$\implies t > \frac{\sigma\sqrt{pln(4pD)}}{D} \tag{40}$$

Hence, whp, the perturbations in each layer $\|U_i\|_2$ are bounded by $\frac{\sigma\sqrt{pln(4pD)}}{D}$ . Plugging this spectral norm bound into Lemma 2 of (Neyshabur et al., 2018), we obtain

$$\max_{x\in\mathcal{X}}|f_{w+u}(x) - f_w(x)| \leq eR\beta^d \sum_i \frac{\|U_i\|_2}{\beta} = eR\beta^{d-1}\sum_i \|U_i\|_2 \tag{41}$$

$$\overset{(i)}{\leq} eR\widetilde{\beta}^d D\frac{\sigma\sqrt{pln(4pD)}}{D} \tag{42}$$

$$= eR\widetilde{\beta}^d\sigma\sqrt{pln(4pD)} \leq \gamma/4 \tag{43}$$

Hence, to satisfy the last inequality, we can choose $\sigma = \frac{\gamma}{eR\widetilde{\beta}^d\sqrt{pln(4pD)}}$

Now, using the above information and the fact that the formula for the KL divergence between two normal distributions which is given by

$KL(\mathcal{N}(w,\Sigma),\mathcal{N}(w_p,\Sigma_p)) = \frac{w-w_p}{\Sigma_p} + tr(\Sigma\Sigma_p^{-1}) - k + ln(\frac{det\Sigma_p}{det\Sigma})$ (Dziugaite & Roy, 2017) , where $k$ is the overall dimensionality, $\Sigma_p$ and $\Sigma$ are the covariance matrices of the prior and posterior, respectively.

We can now evaluate the KL term in Lemma 1 as follows:

$$KL(\mathbf{w}+\mathbf{u}\|\mathbf{P}) = KL(\mathcal{N}(\mathbf{w},\frac{\sigma^2}{D^2}),\mathcal{N}(\mathbf{w_0},\sigma^2))$$

$$= \frac{|\mathbf{w}-\mathbf{w_0}|^2}{\sigma^2} + tr(\frac{\sigma^2/D^2 * I}{\sigma^2 I}) - Dp + ln(\frac{\sigma^2 I}{\frac{\sigma^2}{D^2}I})$$

$$= \sum_{i=1}^{D}(\frac{|\mathbf{w_i}-\mathbf{w_{i0}}|^2}{\sigma^2} + \frac{p}{D^2} + ln(D^{2p}))) - Dp$$

$$\leq \frac{4e^2R^2\widetilde{\beta}^{2d-2}pln(4pD)}{\gamma^2}\sum_{i=1}^{D}\|W_i - W_{i_0}\|_F^2 \leq O(\frac{p}{D} - Dp + 2Dpln(D))$$

$$\leq O\left(\frac{R^2\beta^{2d}pln(4pD)}{\gamma^2}\sum_{i=1}^{D}\frac{\|W_i - W_{i_0}\|_F^2}{\beta^2} + \frac{p}{D} - Dp + 2Dpln(D)\right)$$

$$= O\left(\frac{R^2\prod_{i=1}^{D}\|W_i\|_2^2pln(4pD)}{\gamma^2}\sum_{i=1}^{D}\frac{\|W_i - W_{i_0}\|_F^2}{\|W_i\|_2^2} + \frac{p}{D} - Dp + 2pDln(D)\right)$$

$$\leq O\left(\frac{R^2\prod_{i=1}^{D}\|W_i\|_2^2pln(4pD)}{\gamma^2}\sum_{i=1}^{D}\frac{\|W_i - W_{i_0}\|_F^2}{\|W_i\|_2^2} + \frac{p}{D} + Dp(2ln(D) - 1)\right)$$

Hence, for any $\tilde{\beta}$ such that $|\tilde{\beta} - \beta| \le \frac{\beta}{d}$ for weights $W_i, i = 1, 2.., D$, and weights at initialization $W_{i_0}, i = 1, 2.., D$ with probability $> 1 - \delta$, substituting the above value of $KL(\mathbf{w} + \mathbf{u} \| \mathbf{P})$ into Lemma 1, we obtain

$$\mathbb{P}[arg\ \max_i f(x)_i \ne y] \le \hat{\mathcal{R}}(f) + \sqrt{\frac{\frac{R^2 \prod_{i=1}^D \|W_i\|_2^2 pln(4pD)}{\gamma^2} \sum_{i=1}^D \frac{\|W_i - W_{i_0}\|_F^2}{\|W_i\|_2^2} + \frac{p}{D} + Dp(2ln(D) - 1) + ln(Dm)/\delta}{m}}$$

(44)

This proves equation (13). To obtain the proof of (12), we can just change the distribution of the prior to be $P \sim \mathcal{N}(0, \sigma^2)$ and obtain the KL divergence term

$$KL(\mathbf{w} + \mathbf{u} \| \mathbf{P}) = KL(\mathcal{N}(\mathbf{w}, \frac{\sigma^2}{D^2}), \mathcal{N}(\mathbf{0}, \sigma^2))$$

(45)

$$= \frac{|\mathbf{w}|^2}{\sigma^2} + tr(\frac{\sigma^2/D^2 * I}{\sigma^2 I}) - Dp + ln(\frac{\sigma^2 I}{\frac{\sigma^2}{D^2} I})$$

(46)

Subsequently, following the same steps as in the proof of the previous equation, we obtain the generalization error bound

$$\mathbb{P}[arg\ \max_i f(x)_i \ne y] \le \hat{\mathcal{R}}(f) + \sqrt{\frac{\frac{R^2 \prod_{i=1}^D \|W_i\|_2^2 pln(4pD)}{\gamma^2} \sum_{i=1}^D \frac{\|W_i\|_F^2}{\|W_i\|_2^2} + \frac{p}{D} + Dp(2ln(D) - 1) + ln(Dm)/\delta}{m}}$$

(47)

which also proves equation (12), completing the proof of the whole Theorem.

### A.2.2 Proof of Theorem 3

To prove this theorem, we utilize the following upper bound relating the Rademacher complexity of the margin loss function, with respect to the network's weights, to its covering number (Liao, 2020):

$$\mathfrak{R}_n(\mathcal{G}_\gamma(\mathcal{F}_{\mathcal{D}})) = \inf_{\beta > 0} \left( 4\beta + \frac{12}{\sqrt{m}} \int_\beta^{B/2} \sqrt{log\ \mathcal{N}(\mathcal{G}_\gamma, \epsilon, L_2(S))} \right)$$

(48)

where $\mathcal{N}(\mathcal{G}_\gamma, \epsilon, L_2(S))$ denotes the $\epsilon$-covering of the margin loss function class $\mathcal{G}_\gamma$ with respect to the metric $L_2(S)$, $\mathcal{F}_{\mathcal{D}}$ denotes the function class represented by a $D$ layer neural network, $B = \sup_{f \in \mathcal{F}, x} |g_\gamma(f, x)|$ and $g_\gamma(f_w(x), y) \in \mathcal{G}$ denotes the margin loss function for input $x$ with corresponding label $y$, induced by the network with parameters $w$.

Now, given a network of depth $D$, we can break down the entire network covering number $\mathcal{N}(\mathcal{G}_\gamma, \epsilon, L_2(S))$ into a product of the covering number of the individual layers as follows (Bartlett et al., 2017).

$$\mathcal{N}(\mathcal{G}_\gamma, \epsilon, L_2(S)) \stackrel{(i)}{=} \frac{1}{\gamma} \mathcal{N}(\mathcal{F}, \epsilon, L_2(S)) = \frac{1}{\gamma} \prod_{i=1}^D \mathcal{N}(\{f_i(W_i, x | W_1, W_2, .. W_{i-1}) : W_i \in \mathcal{W}\}, \epsilon_i, \|.\|)$$

(49)

where equality $(i)$ is obtained by taking into account the fact that the Lipschitz constant of the margin loss over the function class $\mathcal{F}$ is $1/\gamma$, $W_i$ denotes the weights of the layer $i$, $f_i(x)$ denotes the output of the $i_{th}$ layer, $\mathcal{W}$ denotes the set of all possible values for the weights of different layers.

Now, supposing the cover radius for the functions in layer $i$ is given by $\epsilon_i$, the error tolerance at layer $i$ is $\epsilon_i$. Thus, since the overall network output can be represented as the function of the $i_{th}$ layer as $f_D(f_{D-1}(..f_i(.)..))$, the error propagated to the final layer is

$$f_D(f_{D-1}(..\epsilon_i)..) = \rho_D \|W_D\|_2 \rho_{D-1} \|W_{D-1}\|_2 .. \epsilon_i = \prod_{j=i+1}^D \rho_j s_j * \epsilon_i$$

(50)

Thus, to ensure that the covering bound of the whole network is $\epsilon$, it should be equal to the error tolerance stated in the equation above. Thus, $\epsilon = \prod_{j=i+1}^{D} \rho_j s_j * \epsilon_i$, or

$$\epsilon_i = \frac{\epsilon}{\prod_{j=i+1}^{D} \rho_j s_j} = \frac{\epsilon}{B_{i+1:D}^{jac}} \tag{51}$$

where $B_{i+1:d}^{jac}$ denotes an upper bound on the Jacobian of the sub-network from the $i_{th}$ to the $d_{th}$ layer.

Now, we use the covering number bound for the individual layers as described above and substitute it into the product of covering bounds in equation (50) to get the covering bound for the whole network:

$$\mathcal{N}(\mathcal{G}_\gamma, \epsilon, L_2(S)) = \frac{1}{\gamma} \prod_{i=1}^{D} \left(1 + \frac{2b_i L_{w_i}}{\epsilon_i}\right)^{2pr} \tag{52}$$

Substituting $\epsilon_i$ into the above equation, we get

$$\mathcal{N}(\mathcal{G}_\gamma, \epsilon, L_2(S)) = \frac{1}{\gamma} \prod_{i=1}^{D} \left(1 + \frac{2b_i L_{w_i} B_{i+1:D}^{jac}}{\epsilon}\right)^{2pr} \tag{53}$$

Furthermore, substituting $L_{w_i} = \rho_i B_{1:i-1}^{jac} R$ from Lemma A3, we get the overall covering number as

$$\mathcal{N}(\mathcal{G}_\gamma, \epsilon, L_2(S)) = log \, \frac{1}{\gamma} \prod_{i=1}^{D} \left(1 + \frac{2b_i B_{1:i-1}^{jac} \rho_i B_{i+1:D}^{jac}}{\epsilon}\right)^{2pr} \tag{54}$$

Substituting the above into equation (49), we obtain the Rademacher complexity for the full network:

$$\hat{\mathfrak{R}}_n(\mathcal{G}_\gamma(\mathcal{F}_D)) = inf_{\beta>0}\left(4\beta + \frac{12}{\sqrt{m}} \int_\beta^{B/2} \sqrt{log \prod_{i=1}^{D} \left(1 + \frac{2b_i B_{1:i}^{jac} \rho_i B_{i+1:D}^{jac} R}{\epsilon}\right)^{2pr} d\epsilon}\right) \tag{55}$$

$$\leq inf_{\beta>0}\left(4\beta + \frac{12}{\sqrt{m}} B/2 \sqrt{log \prod_{i=1}^{D} \left(1 + \frac{2b_i \rho_i B_{\backslash i}^{jac} R}{\beta}\right)^{2pr} d\epsilon}\right) \tag{56}$$

Substituting $\beta = 4B_{1:D}^{jac} R pr \sqrt{2}$, we get

$$\hat{\mathfrak{R}}_n(\mathcal{G}_\gamma(\mathcal{F}_D)) = 16\sqrt{2} R B_{1:D}^{jac} pr + \frac{6B}{\sqrt{m}} \sqrt{log \, \frac{1}{\gamma} \prod_{i}^{D} \left(1 + \frac{2b_i \rho_i B_{\backslash i}^{jac} R}{4B_{1:D}^{jac} R pr \sqrt{2}}\right)^{2pr} d\epsilon} \tag{57}$$

Moreover,

$$B = | \sup_{f \in \mathcal{F}, x} g_\gamma(f, x)| \leq \frac{\| \sup_{f \in \mathcal{F}} f(x)\|_2}{\gamma}$$

$$\leq \frac{\|f_{W_d}(f_{W_{d-1}}(..f_{W_1}(x)..)\|_2}{\gamma} \leq \frac{\|J_{W_d} f_{W_{d-1}}(..f_{W_1}(x)..)\|_2}{\gamma}$$

$$\leq \frac{\|J_{W_d} J_{W_{d-1}}(..f_{W_1}(x)..)\|_2}{\gamma} \leq \frac{\|J_{W_d} J_{W_{d-1}} J_{W_{d-2}}....J_{W_1}\|_2 \|X\|_2}{\gamma}$$

$$\leq \frac{B_{1:D}^{jac} R}{\gamma}$$

where $B_{1:D}^{jac}$ denotes the uniform upper bound on the Jacobian of the network from the $1_{st}$ to the $D_{th}$ layer. After this substitution, many of the terms are canceled, and we obtain:

$$\hat{\mathfrak{R}}_n(\mathcal{G}_\gamma(\mathcal{F}_D)) = 16\sqrt{2}RB_{\backslash i}^{jac}pr + \frac{6B}{\sqrt{m}}\sqrt{log \prod_{i}^{D}\left(1 + \frac{b_i\rho_i B_{\backslash i}^{jac}R}{2B_{1:D}^{jac}Rpr\sqrt{2}}\right)^{2pr} - log\ \gamma} \tag{58}$$

$$\implies \hat{\mathfrak{R}}_n(\mathcal{G}_\gamma(\mathcal{F}_D)) = B_{1:D}^{jac}R\left(16\sqrt{2}pr + \frac{6}{\gamma\sqrt{m}}\sqrt{log \prod_{i}^{D}\left(1 + \frac{b_i}{\|W_i\|_2 2pr\sqrt{2}}\right)^{2pr} - log\ \gamma}\right) \tag{59}$$

Where (59) is obtained from (58) by substituting $B = B_{1:D}^{jac}R/\gamma$ and decomposing the Jacobian of the entire network as $B_{1:D}^{jac} = B_{1:i-1}^{jac}\rho_i\|W_i\|_2 B_{i+1:D}^{jac}$ and canceling the common terms.

Hence, utilizing the inequality $\|W_i\|_F \leq \sqrt{r}\|W_i\|_2$ the second terms inside the brackets in the $log$ (i.e., $\frac{b_i}{2pr\sqrt{2}}$) in the above equation would be of order $O(\sqrt{r}/(2pr \times \sqrt{2})) \approx O(1/(p\sqrt{r}*2\sqrt{2})) < \frac{1}{2\sqrt{2}} << 1$ for sufficiently large $p$ and decrease further below 1 with increasing width of the weight matrices. For instance, for $p = 512$ and supposing a full rank matrix, rank $r = p = 512$, this term would be $1/(1024*\sqrt{2*512}) = 3.0\times10^{-5} << 1$ and is approximately closer to zero.

Now, since the second term inside the product term in the $log$ in equation (59) would be much smaller than 1 for large enough $p$, we would be able to employ the inequality

$$iff\ \ \forall_i\ x_i << 1, x_i \approx 0$$

$$\implies \prod_{i=1}^{D}(1 + x_i) < \sum_{i=1}^{D}(1 + x_i)$$

$$\implies log(\prod_{i=1}^{D}(1 + x_i)) < log(\sum_{i=1}^{D}(1 + x_i))$$

which is a special form of Jensen's inequality. Hence, using this inequality in the above equation, we can convert the product into sums, and can simplify equation (59) as follows:

$$\mathfrak{R}_n(\mathcal{G}_\gamma(\mathcal{F}_D)) = B_{1:d}^{jac}R\left(16\sqrt{2}pr + \frac{6}{\gamma\sqrt{m}}\sqrt{log \prod_{i=1}^{D}\left(1 + \frac{b_i}{\|W_i\|_2 2pr\sqrt{2}}\right)^{2pr} - log\ \gamma}\right)$$

$$\leq B_{1:d}^{jac}R\left(16\sqrt{2}pr + \frac{6}{\gamma\sqrt{m}}\sqrt{log \sum_{i=1}^{D}2pr\left(1 + \frac{b_i}{\|W_i\|_2 2pr\sqrt{2}}\right) - log\ \gamma}\right)$$

$$\leq B_{1:d}^{jac}R\left(16\sqrt{2}pr + \frac{6}{\gamma\sqrt{m}}\sqrt{log \sum_{i=1}^{D}\left(2pr + \frac{b_i}{\|W_i\|_2\sqrt{2}}\right) - log\ \gamma}\right)$$

Substituting the value of $b_i = \|W_i\|_F$, the proof of the Theorem is completed.

