# OpenReview forum: "Nonvacuous Generalization Bounds For Deep Networks With Improved Size Dependence"
_TMLR — Rejected by TMLR_

### Review · Reviewer_aASv · 2025-10-13

**Summary Of Contributions:**

The paper considers the problem of using deep neural networks for the class of multiclass classification. New generalization bounds are proven which are stronger compared to older results in the deep learning theory literature. At the core of the improvements lie two new bounds on the covering number of function classes that are learnable by neural networks. The first bound is proven under the assumption that the weights of the neural network do not change much compared to the values they have at initialization, whereas the second involves the additional assumption that the norm of the weight vector corresponding to each hidden unit is upper-bounded by a constant. The second result specifically demonstrates that the additional assumption can lead to an exponential improvement in the implied bounds. The theoretical results are complemented by experimental evaluation.

I believe the main strength of the paper lies in the results, which I consider to be important and interesting, due to the improvements over prior work that they represent. The assumptions involved in their proofs are reasonable within the context of the overall literature, so I don't have any concerns there. Additionally, the writing is generally good and easy to follow (I have some minor comments which are given later in the requested changes).

Overall, I think this is a good paper and I recommend acceptance.

**Audience:**

Yes

**Audience Explanation:**

The paper is about generalization in deep learning, which is a topic that I believe lies at the core of the TMLR community.

**Broader Impact Concerns:**

I do not have any concerns about the ethical implications of the work.

**Claims And Evidence:**

Yes

**Claims Explanation:**

All the main results are supported by proofs which, to the best of my understanding, are correct. Additionally, I do not have any concerns about the experimental evaluation.

**Requested Changes:**

I have a number of comments about some minor edits that I'd like to see.

The first has to do with the commentary that follows Theorem 3. It is argued that $\|W_i\|_2$ is upper-bounded roughly by $\sqrt{p}$. This is used to interpret the bound given earlier. However, for the analysis to make sense, we'd want a lower bound on on $\|W_i\|_2$. Of course, this has nothing to do with the correctness of the theorem, and I understand the point the authors are trying to illustrate. However, I think it'd be better if the description had been presented in a slightly less sloppy fashion.

The second has to do with the organization of the appendix. I don't have any concerns about correctness, but the order in which the material was presented didn't seem natural to me in some places. The appendix starts with a couple of lemmas (A1 and A2) which are proven in this work (but the proofs are given later), followed by a lemma from an older paper (Lemma A3). Lemma A3 is proven right after stated, which is not really necessary, given that it's a known result. Then, Appendix A.2 starts with the comment ``Before describing their proofs, we introduce an additional lemma which will be useful in the analysis of covering numbers''. The lemma in question is Lemma 1 from the main body. Thus, rather than introducing a new result, a proof from a result from the main body is given. Then, the proofs of lemmas A1 and A2 are given.

I think a better way to organize the contents of the appendix would be to start with Lemma A3 (since it's older work -- with or without proof, it doesn't make much of a difference), then proceed with Lemmas A1 and A2 with their proofs, and then give the proof of Lemma 1 and all the other results from the main body. Alternatively, Lemma A2 could be given after Lemma 1, given that it's not really invoked in its proof.

Finally, I have one very minor comment about Table 1. A parenthesis is missing in one of the bounds given in the second-to-last row of the table. Also, I assume the authors use the notation $\widehat{\mathcal{O}}$ as an alternative to $\widetilde{\mathcal{O}}$. It may be good to add a comment about that in Section 1.2, since I don't think this notation is standard.

Overall, I think this is a good paper, and I don't consider the issues with the presentation that I pointed out to above to be serious by any means.

---

> ### Author Response · Authors · 2025-10-30
> **Corrections of the changes requested by the reviewer**
>
> Dear reviewer,
> I thank you for your careful reading and review of my paper and for providing constructive comments and criticisms. Your valuable feedback has allowed me to improve the overall clarity and presentation of my work.
>
> Based on your feedback, here is a summary of the changes I have incorporated:
>
> 1) The Lemma A3 which is a variant of an older lemma presented in Bartlett. et. al and is a precursor to Lemma 1 is now placed at the top in Appendix. I also decided to keep its proof since it is now slightly modified to accomodate the request changes of other reviewers as well.
> This Lemma is then followed by Lemmas A1 and A2 (which are now Lemmas A2 and A3 in the revised manuscript) along with their proofs stated right after introducing them. After introducing the new lemmas, the proof of the ones from the main body is derived.
>
> 2) The Table 1 has been corrected by including the missing parenthesis and replacing $\widehat{\mathcal{O}}$ with the corrected notation $\widetilde{\mathcal{O}}$.
>
> 3) The upper bound on $\|W_i\|_2$ is now removed. I could have replaced it with the lower bound as $O(\sqrt{p})$ but it would have then led to $\prod_d \|W_d\|$ (or $B^jac$) growing exponentially with depth which would have multiplied an additional factor of D log p into the bound. Since, I am now placing constraints on $B^jac$  to avoid this based on the comments of reviewers, this scaling factor in the denominator has been ignored.
>
> I hope these amendments and corrections do address your main concerns regarding the paper and I look forward to your further comments on the revised manuscript. Thank you for taking your valuable time to review my manuscript.

---

> > ### Comment · Action_Editor_CpVn · 2025-11-21
> >
> > Dear Reviewer,
> >
> > Many thanks for your participation in the reviewing process.
> > Could you please provide your final recommendation so I can move forward with the final decision?
> >
> > Best regards,
> >
> > AE

---

> > > ### Comment · Action_Editor_CpVn · 2025-11-25
> > >
> > > Dear Reviewer,
> > >
> > > I cannot move on with the decision without your final recommendation. Would you mind updating your final recommendation?
> > >
> > >
> > > Alternatively, if you have something to add to the discussion, please feel free to interact with me, the other reviewers or the authors.
> > >
> > >
> > > Many thanks for your contribution to TMLR.
> > >
> > > Best regards,
> > >
> > > AE

---

### Review · Reviewer_3h1F · 2025-10-18

**Summary Of Contributions:**

**Summary**: This paper proposes tighter norm-based generalization bounds that scale better with neural network's architectural complexity than previous work such as [Bartlett'17], [Neyshabur'15, 18] and [Golowich'18].

**Audience:**

Yes

**Audience Explanation:**

Neural networks have notoriously known to have mostly vacuous generalization bounds because of their size. Therefore, the discoveries of techniques to provide non-vacuous guarantees are highly appreciated in the community.

**Broader Impact Concerns:**

This work is purely theoretical and I cannot foresee any immediate societal impact.

**Claims And Evidence:**

No

**Claims Explanation:**

**Comments**: I am mainly concerned with the correctness of the main theorem on the general case (Theorem 2). As far as I understand, the authors used the same method as [Bartlett'17] to break the neural network's cover into $D$ covers for each layer (using Lemma A.7 as in [Bartlett'17] or Lemma A3 as in the author's manuscript).

For the $i^\mathrm{th}$ layer, the author set the granularity of the cover to $\epsilon_i=\frac{\epsilon}{\rho_i\prod_{j>i}\rho_js_j}$ and claims that the final granularity of the cover for the entire network is $\epsilon$. However, by Bartlett's Lemma A.7 (or author's Lemma A3), the final granularity of the entire network is:

$$
\delta=\sum_{i=1}^D \epsilon_i \rho_i \prod_{j>i}\rho_j s_j = \sum_{i=1}^D \frac{\epsilon}{\rho_i\prod_{j>i}\rho_js_j}\rho_i \prod_{j>i}\rho_j s_j = D\epsilon.
$$

Therefore, the cover in the proof of Lemma 1 has a granularity of $D\epsilon$, not $\epsilon$ as authors claimed. As a result, the final $\epsilon$-covering number bound (after correctly applying Lemma A3) of the entire network (proof of Lemma 1 in Appendix A.2.1) should have an additional multiplicative term of $D^2$ in the **non-logarithmic** term. Therefore, Theorem 2 will have a linear dependency on the depth of the neural networks. The supposed tighter result is due to the fact that the authors did not normalize the layer-wise covering number's granularity. In [Bartlett'17], this is done by setting $\epsilon_i=\frac{\alpha_i\epsilon}{\rho_i\prod_{j>i}\rho_j s_j}$ where $\sum_{j=1}^D \alpha_j=1$.

In other words, unless the authors can justify that the resolution of the covering number in their Lemma 1's proof is indeed $\epsilon$, their final bound has a linear dependency on $D$, the network's depth, which contradicts with the author's original claim that "We propose a novel generalization bound that is nonvacuous and has a better dependence on the network dimensions (depth and width)".

**Minor comments**: The authors should be more precise with classic learning theory results. For example:

1. In Eqn. (6) (and many equations that follows), the additive logarithmic term of $\delta$ is $\sqrt{\frac{\ln 1/\delta}{n}}$, not $\sqrt{\ln\frac{1/\delta}{n}}$.
2. In Eqn. (12), I suppose that the authors are referring to the Dudley's entropy bound, which is actually an integral over the covering number's granularities, not just a square-root of the covering number with a random granularity $\epsilon_i$. Also, the bound applies for the empirical Rademacher complexity rather than the expected Rademacher complexity.
3. In Eqn. (13), I assume that thhe authors are using the Pollard bound, which is an upper bound on the empirical Rademacher complexity, not the definition of empirical Rademacher complexity itself.
4. In Lemma A3, the granularity of the entire network should be $\delta=\sum_{j\le L}\epsilon_j\rho_j\prod_{l=j+1}^L\rho_lc_l$ I suppose (if we are following [Bartlett'17]), not $\delta=\sum_{j\le L}c_j\rho_j\prod_{l=j+1}^L\rho_lc_l$. Otherwise, the precision of layer-wise covers have no bearings on the overall network's cover.
5. (Minor) In the proof for Lemma 1, the authors forgot $\ln(W^2)$ at some last steps.

**Requested Changes:**

I would appreciate it if the authors improve the current manuscript by:

1. Elaborate further on the correctness of the main theorem (Theorem 2).
2. Revise the classic results in learning theory and polish them more precisely (for example, Dudley and Pollard bound should definitely be fixed).

---

> ### Author Response · Authors · 2025-10-30
> **Corrections of proof to Theorem 2 and addressing other minor comments.**
>
> Dear reviewer,
> I thank you for your careful reading and review of my paper and for providing constructive comments and criticisms. Your valuable feedback has allowed me to improve the overall clarity and presentation of my work.
>
> Based on your feedback, here is a summary of the changes I have incorporated:
>
> I have adjusted Lemma A3 (which is now Lemma A1 in revised manuscript) and added normalization terms $\alpha_i$ to the granularities of the layerwise coverings in Lemma 1, which is a precusor to the main Theorem 2 on which you raised concerns. Moreover, the constant factors in the granularities of each covering have been adjusted such that the overall covering of the whole network is now bounded by $\epsilon$ while keeping the results of the theorem the same. Moreover, I have addressed the minor issues raised by you as follows:
> 1) In all the bounds, I have replaced the erroneous term $\sqrt{\ln \frac{c/\delta}{n}}$  with $\sqrt{\frac{\ln c/\delta}{n}}$
> 2) In equation (12), I have applied the correct but slightly modified form of the Dudley entropy integral (derived in Lemma A.8 of Bartlett et. al, 2017) and introduced correct constants 4 and 12 as well to make sure that the upper bound is an integral over the square root of covering number, and not based on the vanilla covering number.
> 3) Similarly, in equation (13) I have correctly specified the Dudley entropy integral with the correct constants 4 and 12 and mentioned it as an upper bound of the Empirical Rademacher complexity instead of equating the integral with it.
> 4) In proof of Lemma A1 (originally Lemma A3), I have corrected $\delta = \frac{1}{D}\sum_{j\leq D}\rho_{j}c_{j}\prod_{l=j+1}^{D}\rho_{l}c_{l}$ wirh $\delta = \frac{1}{D}\sum_{j\leq D}\rho_{j}\epsilon_{j}\prod_{l=j+1}^{D}\rho_{l}c_{l}$
> 5) In the proof of Lemma 1, the omitted $ln (W^2)$ terms have now been added.
>
> I hope these amendments and corrections address your main concerns regarding the paper and I look forward to your further comments on the revised manuscript. Thank you for taking your valuable time to review the manuscript.

---

> ### Comment · Reviewer_3h1F · 2025-10-31
> **Questions for Lemma A1**
>
> Dear Authors,
>
> Thank you very much for taking the time to correct the manuscript and review some of the small errors I pointed out. However, I still find your modified proof a bit odd, specifically where you suddenly have $1/D$ multiplicative factor in the global covering granularity.
>
> Specifically, in your revised manuscript, the way that you remedied the dependency of $D^2$ in the final covering granularity is that you divide the Lipschitz constant of every layer by $D$ then re-parameterize the weight matrices by multiplying them with $D$:
>
> $$
> \rho_i(W_ix+b_i) = \frac{\rho_i}{D}(DW_ix+Db_i).
> $$
>
> This re-parameterization is **not** invariant as the authors claimed because for each layer $i$, $W_i\in\mathcal{B}\_i$ and therefore, $DW\_i\in D\mathcal{B}\_i$. For example, if $\mathcal{B}\_i$ is a norm ball then $DW\_i$ belongs to the "dilated" norm ball by a factor of $D$. In other words, by rescaling the $\rho\_i$'s, you end up having to cover some parameters spaces that is $D$ times larger in radius than the original ones. Therefore, there is actually no tightness achieved by rescaling the Lipschitz constant. The overall cover granularity is still $\sum\_{j\le D}\rho\_jc\_j\prod\_{\ell=j+1}^D \rho\_\ell c\_\ell$, without $D^{-1}$ that the authors added.
>
> For these reasons, the modified steps from Eqn. (33) and beyond is still not justified by the modified Lemma A.1. provided.

---

> ### Author Response · Authors · 2025-11-03
> **Change in the derivation of Theorem 2**
>
> Dear Reviewer,
>
> Thanks you very much for your taking your time to provide feedback on the manuscript and specifically on Theorem 2. To address your concerns, I have adapted my approach to prove the theorem by using a PAC-Bayesian framework over a perturbation bound on the network weights.
>
> More specifically, I used a modified form of the technique used by Neybashur et. at, 2018 to derive a posterior distribution over the perturbed predictor $f_\mathbf{w+u}$ such that the perturbations in each layer satisfy an upper bound. My version of the proof deviates by assuming a different variance of $\mathcal{N}(0,\sigma^2/D^2)$, than the ones used by the authors $\mathcal{N}(0,\sigma^2)$ based on the observation that $\|U_i\|_2 \leq 1/d \|W_i\|_2$. Subsequently, using this posterior distribution I derive a different $KL(w+u\|P)$ term which has better depth scaling than proposed by the authors Neybashur et. at, 2018. However, the proposition of the theorem is now slightly different than stated in the previous manuscript but with a similar growth rate.
>
> With this change, I hope you find this proof addressing some of your major concerns about the proof of the Theorem. I thank you again for your valuable time on reviewing the manuscript and look forward to any further comments or queries you might have regarding the revised manuscript.

---

> > ### Comment · Reviewer_3h1F · 2025-12-06
> > **Response to PAC-Bayes Proof, Apologies for Delayed Response**
> >
> > Dear Authors,
> >
> > I really appreciate the authors response and the revision to the manuscript. My sincere apologies for the delayed response because of an unforseen circumstance with my health. I hope this response is not too late.
> >
> > I have read through your revised proof of Theorem 2 using Neyshabur's PAC-Bayes framework and I think the idea looks correct. By selecting the Gaussian prior with a standard devision $D$ times smaller than Neyshabur's original work, you seemed to successfully ``deferred" the expensive depth dependency to log-terms.
> >
> > Nonetheless, I have to admit that I am not too well-versed in PAC-Bayes literature. Therefore, my confidence on the assessment of the proof is low as I have not found the time to check if selecting such a prior violates some other weird conditions enforced by the framework.
> >
> > Despite being convinced by your revised proof, I am still really reluctant to recommend acceptance after reading through the Action Editor's comments. However, I really do encourage the authors to revise the manuscript once more. Once again, my apologies for not being able to allocate more time to check your work more thoroughly.

---

> ### Comment · Action_Editor_CpVn · 2025-12-06
>
> Dear Reviewer 3h1F, dear Authors,
>
>
> Thank you for making efforts to continue reading the paper to give as thorough feedback to the authors as possible so they can achieve closure and keep learning! I really appreciate that you were the only reviewer who caught the serious errors (a testament of your academic capabilities and the fact you are taking the reviewing task seriously), and the fact that you are still trying to help the authors despite the errors (a testament of your kindness).
>
>
> However, I have to disagree with your opinion on the correctness of Theorem 2. As I said before, I was very suspicious of it for the following reasons:
>
> 1. the authors provided an incorrect proof the same result from a frequentist perspective.  PAC Bayesian bounds and their frequentist counterparts are more similar than appear at first glance (like dialects of a proof language): The KL divergence in Lemma 1 in [4] plays a similar role to covering number argument in [5]. If the PAC Bayes bound works, it should be possible to modify the frequentist proof to work as well.
> 2. Just as the frequentist proof closely followed [5] very closely and claimed to tighten the bound with a very simple scaling argument, the new proof follows [4] very closely in almost all respects except that the authors rescale the variance of the posterior by a factor of 1/D^2. In that respect, it is not such a fundamentally different attempt form the previous frequentist attempt. Overall, **I would strongly urge the authors to be very suspicious of any proof they attempt which appears to achieve a far superior result to an extremely famous paper with almost exactly the same proof up to a simple scaling modification**: if such a simple modification could improve the result, why would the authors of the original paper not have done it?
>
>
> More concretely, **the error is in the calculation of the KL divergence**.
>
>
> On page 24 of the current submission, the authors incorrectly use equation (1) from [DR] to calculate the KL divergence between the prior on $w$ (variance $\sigma^2$) and the posterior (variance $\sigma^2/d$ in the current submission).
>
>
> The correct equation (see the paper) is
>
> $$KL(N_q,N_p)=\frac{1}{2}(TR(\Sigma_p^{-1}\Sigma_q)-k +(\mu_p-\mu_q)^\top \Sigma_p^{-1}(\mu_p-\mu_q)+\log(\frac{\det(\Sigma_q)}{\det(\Sigma_p)}))$$
>
> **Here $k$ is the dimensionality of the whole prior/posterior. In this case, that is $Dp^2$, not $Dp$ as claimed in the submission (the weights live in $\mathbb{R}^{p\times p}$ not in $\mathbb{R}^p$).** Many errors arise out of this:
>
> 1. (*Actually makes the bound pessimistic*) First, the term $-k$ should be $-Dp^2$ instead of $-Dp$.
> 2. (*Makes the bound smaller than it should be and therefore incorrect, but would mostly just change the final result*). The term $TR(\Sigma_p^{-1}\Sigma_q)$ is incorrectly calculated as $1/D^2 pD=\frac{p}{D}$ (after the sum), when it should be $[1/D^2] p^2D=\frac{p^2}{d}$.
> 3. (**FATAL**) The term $\log(\frac{\det(\Sigma_q)}{\det(\Sigma_p)}))$ is calculated as $\log(D^{2p})$ ($\log(D^{2pD})$ if we do the whole calculation over every layer at the same time)  when it should be $\log(D^{2p^2D})$ (taking all layers at the same time. This means that the additive term $Dp(2\log(D)-1)$ in the current paper should instead be $Dp^2 (2\log(D)-1)$. Note that this term on its own already makes the bound **comparable to naive parameter counting** [LS, Graf] in the case where the spectral norms are all equal to $1$ apart from log terms (the argument of the log is $D$ versus $Dp$ in paracount (norms 1)). If there is any new result which can be proved here, it will certainly not be nearly as strong as claimed.
>
>
> Another minor point, in the paper the term $(\mu_p-\mu_q)^\top \Sigma_p^{-1}(\mu_p-\mu_q)$ is first incorrectly written as $\frac{w-w_p}{\Sigma_p}$ in the preamble before the actual sequence of equations: there is a square norm missing at the top and "dividing by a matrix" is unusual notation at best. Fortunately later in the proof the squared norm reappears and the term becomes $\frac{\|w-w_p\|^2}{\sigma^2}$ which is correct.
>
>
>
>
>
>
>
> [4] Behnam Neyshabur, Srinadh Bhojanapalli, and Nathan Srebro. A pac-bayesian approach to spectrally- normalized margin bounds for neural networks, 2018 (ICLR)
>
> [5] Bartlett et al. Nearly-tight vc- dimension and pseudodimension bounds for piecewise linear neural networks. JMLR 19.
>
>
> [DR] Dziugaite and Roy. Computing Nonvacuous Generalization Bounds for Deep (Stochastic) Neural Networks with Many More Parameters than Training Data
>
>
> [LS] Long and Sedhi. Generalization bounds for deep convolutional neural networks
>
> [Graf] Graf et al. On the excess capacity of Neural Networks.

---

### Review · Reviewer_VX1n · 2025-10-20

**Summary Of Contributions:**

This manuscript examines the generalization ability of DNNs despite their substantial overparameterization. It introduces novel nonvacuous generalization bounds that yield tighter and more realistic estimates of the Rademacher complexity of deep networks. The main contribution lies in a refined analysis of the covering number, leading to a significantly milder dependence on network depth, with the derived bounds scaling as $O(\sqrt{Dpr})$. Moreover, under reasonable assumptions on weight norms, the analysis establishes a sub-logarithmic depth dependence of $O(\sqrt{\log D})$, representing a marked improvement over previous polynomial-depth bounds. Extensive empirical evaluations further confirm that the proposed bounds consistently provide tighter estimates across various architectures and datasets.

**Audience:**

Yes

**Audience Explanation:**

This manuscript investigates the generalization capability of DNNs despite their substantial overparameterization, which has been largely neglected in prior research.

**Claims And Evidence:**

Yes

**Claims Explanation:**

Extensive empirical evaluations confirm that the proposed bounds consistently provide tighter estimates across various architectures and datasets.

**Requested Changes:**

- Although the experimental design and implementation are sound, the empirical evaluation remains confined to VGG-style convolutional architectures and the CIFAR-10 dataset. As a result, the generalization behavior of deeper or more contemporary architectures (e.g., ResNets or Transformers) and larger, more complex datasets like ImageNet has not been explored. This limited empirical scope constrains the extent to which the proposed bounds can be generalized to modern large-scale models and tasks.
- The derivation of the sub-logarithmic bound relies on the assumption that each hidden unit’s weight norm is upper-bounded by one. While this assumption simplifies the theoretical analysis and ensures tractability, it may not accurately reflect the properties of practical large-scale neural networks, where weight norms often vary considerably. Consequently, this simplification could restrict the practical applicability and interpretive strength of the derived bound.
- The presented theoretical framework does not explicitly incorporate the influence of optimization dynamics, particularly the implicit regularization effects introduced by stochastic gradient descent. Although the authors acknowledge this omission in the *Future Directions* section, a more explicit theoretical discussion or controlled empirical study examining the impact of stochastic gradient descent on generalization would have provided a more comprehensive understanding and strengthened the overall contribution.
- Despite the manuscript’s mathematical rigor and formal clarity, the exposition could benefit from more intuitive explanations or geometric interpretations that clarify why the proposed bound exhibits a milder dependence on network depth. Without such interpretive insights, readers who are less familiar with covering number theory or Rademacher complexity analysis may find the derivations abstract and difficult to connect to practical implications.

---

> ### Author Response · Authors · 2025-10-30
> **Added more extensive validation of bounds and relaxing restrictions.**
>
> Dear reviewer,
> I thank you for your careful reading and review of my paper and for providing constructive comments and criticisms. Your valuable feedback has allowed me to improve the overall clarity and presentation of my work.
>
> Based on your feedback, here is a summary of the changes I have incorporated:
>
> 1) I have validated the effectiveness bounds on different architectures including Residual Networks of different depths and have performed the experiments and validated the tightness of our bound on more diverse datasets apart from CIFAR-10 such as CIFAR-100 and Tiny-Imagenet.
> 2) I have adapted the derivation of the tighter version of my bound which scales logarithmically with depth by removing the conditions that the norms of each hidden unit be bounded by 1 and replacing it with the only condition that the width of the network be bounded. As such, I have enlarged the scope of the new bound to cover the generalization behaviour of deep networks in more general conditions where the weight norms can vary significantly from 1.
> 3) I am working on incorporating the influence of the optimization dynamics of stochastic gradient descent (SGD) into the derivation of the bounds. The new bounds including the effects of the optimization algorithm will be reflected in the updated manuscript.
> 4) I have clearly explicated the definitions of technical terms like covering number and Rademacher complexity with examples and illustrations so that these concepts becomes more concise with readers who are unfamiliar with them. I have also included pictures and elaborated on how I derive the bounds of the covering number by relating it to the volume of the function spaces and spheres, which I think will appeal more to the readers.
>
> I hope these amendments and corrections address your main concerns regarding the paper. If you have any further query, please let me know. I look forward to your further comments on the revised manuscript. Thank you for taking your valuable time to review the manuscript.

---

### Comment · Action_Editor_CpVn · 2025-10-28
**Discussion, claims of errors by reviewer 3h1F**

Dear reviewers, dear authors,

All reviews are now available, and there is some disparity between the comments. Please discuss the issues below and the ones mentioned by reviewer 3h1F and the other reviewers.

In particular, reviewer 3h1F has identified what appears to be very **serious correctness issues**: one of the claimed contributions of the paper is the reduction in the dependence on the depth in Theorem 2 compared to the main theorem in [1].

After looking through the proof myself, it appears this is indeed an error: **equation (26) doesn't hold, since the granularity of the global cover has not been adapted**. The proof of Theorem 3 also has the same problem in equation (39).



Reviewer 3h1F also mentions plenty of small inconsistencies in the calculations and formatting. In addition to that, I would like to ask the the authors to the following minor-to-medium points, which potentially affect constants/log terms:

1. As the reviewer mentioned, the integral is missing in equation (13). If you mean the Dudley version, you are also **missing two constants** of 4 and 12 respectively, cf. Lemma A.5 in [1]. Same problem in equation (38).

2. Your lemma A1 is vaguely stated as it appears to relate to a single layer and doesn't need references to the full network. Furthermore, it doesn't hold for $\epsilon_i > 2b_i L_{W_I}$ (because the covering number needs to be greater than 1), so there is at least a minor error here (I believe you are misquoting Vershynin [5] near the middle of your page 20: if you meant to use corollary 4.2.13 page 78, the basis of the exponential should be $1+\frac{2}{\epsilon}$). Furthermore, this is a standard parameter counting argument, so it is surprising that you have a factor of 2 after taking the logs (from the second power of $\epsilon_i$ in the denominator). In future revisions, I would recommend stating the result in a concise notation that doesn't refer to the full network. See for instance Proposition F1 in [2].

3. Similarly, in Lemma A.12, you should explain more clearly that by the "Jacobian being given by $B^{jac}$, you mean that there is a **uniform upper bound** on the Jacobian.

4. At the end of page 18, you use lemma 3.2 from [1] but I think you are missing a ceiling function introducing at least minor errors.

5. You mention that your bounds only scale like $O(\sqrt{Dpr})$, but there are assumptions here: first of all, this rate omits logarithmic factors (e.g. $\log(W^2)$), which should be mentioned. This is certainly something which needs to be said explicitly.
In addition, I would be more careful in the discussion of how the bounds scale in terms of the assumptions on the norms. It appears that you are making these statements based on the assumption that all the spectral norms are exactly equal to 1. If you were instead to assume that the spectral norms are bounded by a constant $C$, then the factor $\log(\prod_i \|W^i\|_2)\leq \log(C^D)=D \log(C)$ (which is present in [4]) **introduces an additional nonlogarithmic factor of $\sqrt{D}$.** (this explains the discrepancy between the rates stated in [4] and [2]). This is actually something which could theoretically play to your advantage if you were to avoid a logarithmic factor in $\prod \|W_i\|$ in your final bound even after correcting all the errors (though it might not be possible).





References:

[1] Bartlett et al. Spectrally normalized margin bounds for Neural Networks.

[2] Ledent et al. Generalization bounds for rank-sparse neural networks. NeurIPS 2025.

[3] On Generalization Bounds for Neural Networks with Low Rank Layers . Pinto, Rangamani and Poggio. ALT 2025

[4] Li et al. On Tighter Generalization Bounds for Deep Neural Networks: CNNs, ResNets, and Beyond

[5] Vershynin, High Dimentional Probability.

---

> ### Author Response · Authors · 2025-11-08
> **Response to Reviewer comments**
>
> Dear reviewer,
>
> Thank you so much for taking your time to review and provide valuable feedback on my paper and for pointing out the issues. Based on your comments on the issues with the proof of Theorem 2, I have provided a new corrected proof based on the PAC-Bayesian framework wherein I use a bound on the perturbation of the weights to derive a novel generalization bounds with lower growth rate. Also, in the proof of Theorem 3, I have now adapted the granularity of the layerwise cover to obtain a global cover of $\epsilon$ (steps illustrated in equations (51,52)).
>
> Also, based on your comments on other issues, here are the changes I have made:
> 1) Corrected the bounds on Rademacher complexity using the Dudley entropy integral and incorporated the constants 4 and 12 as described in its original form [1] in equations (14) and (15).
> 2) Lemma A1 (which is now Lemma A2 in the revised manuscript) does not make any references to the full network and only makes assumptions and derives equations based on the variables for the $i_{th}$ layer itself. Moreover, I have incorporated the correct form of the equation quoted in Vershynin [2] and replaced $(2b_i/\epsilon)$ with the corrected form $(1+2b_i/\epsilon)$ and used this to derive the bound further. Lastly, to get rid of the extra $r^2$ term in $D_{eff} = 2pr-r^2$, which should be the actual number of free parameters, I assumed $2b_i/\epsilon_i>1$ in an attempt to explain the effective dimensionality of each layer, which cannot just be equal to $pr$ without any justification (for instance, it is assumed as h=Dpr in Lemma 2 of [3] in Appendix without any justification and without taking the reparameterization into context and they also reach at a similar result). If you find it inconsistent with the whole derivation, I can simply use $D_{eff} = pr$, referring to Lemma 2 of [3], and ignore the extra steps to avoid making this assumption.
> 3) In the proof of Lemma A2 (now A3) and other theorems as well I have specified that $B^{jac}_{1:D}$ (and other such constants) denotes the uniform upper bound on the Jacobian of the network.
> 4) Since, I have changed the proof of Theorem 2 (to which this comment was related to) from a covering number bound to a PAC-Bayesian bound, the derivations have changed and this is now irrelevant.
> 5) At all places where I compare the growth rate of the different bounds, I have now explicitly mentioned that these rates hold based on the assumption that either the spectral norm of each layer is bounded by 1 (i.e., $\|W_d\|_2<1$) or the product of spectral norms is upper bounded by a constant term (which is independent of network size). Also, I have stated in the comparison that the other factors (such as $\frac{\|X\|_2}{\gamma\sqrt{m}}$) are common and hence ignored. Moreover, I have further stated that the terms appearing due to $ln W^2$ are also ignored due to their recurrent appearance (or that of similar terms) in other bounds as well. Similar arguments were used in the earlier papers such as [3,4] so I think these assumptions are justified.
>
> With these amendments, I hope that most of your concerns are addressed. If you have further queries or comments, please let me know. I thank you again for your valuable feedback and I look forward to your further comments on the manuscript.
>
> References:
>
> [1] Renjie Liao. Notes on rademacher complexity, 2020
>
> [2] Roman Vershynin. High-dimensional probability. 2018
>
> [3] Xingguo Li, Junwei Lu, Zhaoran Wang, Jarvis Haupt, and Tuo Zhao. On tighter generalization bound for
> deep neural networks: Cnns, resnets, and beyond, 2018
>
> [4] Behnam Neyshabur, Srinadh Bhojanapalli, and Nathan Srebro. A pac-bayesian approach to spectrally-
> normalized margin bounds for neural networks, 2018

---

> > ### Author Response · Authors · 2025-11-21
> > **Requesting update on the review process**
> >
> > Dear Reviewers and Action Editors,
> >
> > Thank you so much for taking the time to review the manuscript and for providing me with valuable feedback. Your input has been instrumental in improving the manuscript.
> >
> > Since four weeks have passed since all the reviews were submitted, I would appreciate an update on the current stage of the review process.
> >
> > Also, please let me know if you have any comments regarding the latest changes I made in the paper to address the reviewers' comments.
> >
> > I once again thank you for your valuable support and time and look forward to your further comments.
> >
> > Thanks

---

> ### Comment · Action_Editor_CpVn · 2025-11-21
>
> Dear Authors,
>
>
>
> Many thanks for your patience with the review process.
> It is approximately time for the reviewers to make any final comments and make their "recommendations" (@Reviewer aASv, could you please provide your final recommendation?). You should hear back in the next weeks.
>
> After reading through the updated version again, I am really sorry to say that I still feel the work is in a very early stage and far from publishable since there are still significant errors. I aim to give you more detailed comments together with the final meta review but in the mean time, a quick couple of comments. You should also try to read my previous comments and those of reviewer 3h1F in more detail.
>
> In our previous conversation, I mentioned the issue of parametric covering number bound missing an additive term of $1$ inside the bracket. You have partially addressed this but you didn't check how the change propagates through the network: above equation (30), **you state that "we know the radius of the cover should be comparatively very small to that of the function class**". You shouldn't rely on such vague statements in a rigorous proof. Here, the granularity of the cover depends on the spectral norms of all the other layers, so I don't think one can trivially wave this issue away.
>
> More importantly, there is the issue of comparison to other bounds based on how you interpret logarithmic factors (point 5). I respectfully disagree with your answer. You claim that treating all the norms as constants is standard practice in [3,4]. That is definitely not true of [4], and not quite true of [3] either. In [3], the authors explicitly caution that their sample complexity of Dpr is only valid under the assumption $\|W_d\|=1$ for all $d$ (there are two columns in the table, because the authors are aware and honestly disclose the fact that this is a strong assumption). Note that **there is equality here, not a strict inequality $\|W_d\|<1$ as you assume**: the equality constraint in [3] is a very strong assumption that can still make sense (for instance, if the layers are normalised by spectral norm), your inequality constraint, on the other hand, is not reasonable. This is because if the spectral norm of each layer is small, then *the margin will be very small*, making the bound meaningless. For instance, if you assume that $\|W_d\|\leq 0.99999$, **the maximum achievable margin will decay exponentially with $D$**, which you can use to offset other depth dependency elsewhere in the bound (because you are still expressing the bound in terms of the margin achievable with these conditions).  This is why the argument in the proof of Theorem 3 doesn't make sense: your rate isn't really $\sqrt{\log(D)}$ in any meaningful sense. (related to this, please see the reviewer's comment https://openreview.net/forum?id=aSv29Xh81q&noteId=xkA51Yhk31).
>
> Note that **your Theorem 3 claims that the sample complexity of neural networks is logarithmic in depth**. This is an outlandish claim which can only hold under overly restrictive "assumptions" that obfuscate the true sample complexity. The VC dimension *lower bound* in [5] is quadratic in depth, which matches the rate in both [3] and [6,7] if we impose a constraint of the form $\|W_d\|\leq 1+c$ for small $c$ and consider only the fully parametric (and for [7], full rank) case.
>
> Unfortunately, I haven't checked all the details of your entirely new PAC Bayesian proof of Theorem 2, but I am suspicious of it since it looks very similar to the proof in [4] but achieves a supposedly stronger rate (similarly to the first, purely frequentist submission which was incorrect as well). I will see if I can include more details in the final meta review. I sympathise with the authors: getting familiar with learning theory is a very difficult  and unforgiving task, and I am sure they have already learnt quite a few things writing this draft, but there is still some way to go before publication, in my opinion.
>
> Best wishes,
>
> AE
>
>
>
>
>
> [1] Renjie Liao. Notes on rademacher complexity, 2020
>
> [2] Roman Vershynin. High-dimensional probability. 2018
>
> [3] Xingguo Li, Junwei Lu, Zhaoran Wang, Jarvis Haupt, and Tuo Zhao. On tighter generalization bound for deep neural networks: Cnns, resnets, and beyond, 2018 (preprint)
>
> [4] Behnam Neyshabur, Srinadh Bhojanapalli, and Nathan Srebro. A pac-bayesian approach to spectrally- normalized margin bounds for neural networks, 2018 (ICLR)
>
>
> [5] Bartlett et al. Nearly-tight vc- dimension and pseudodimension bounds for piecewise linear neural networks. JMLR 19.
>
> [6] Ledent et al. Generalization Bounds for Rank-sparse Neural Networks. NeurIPS 2025.
>
> [7] Graf. et al. On Measuring Excess Capacity in Neural Networks. NeurIPS 2022.

---

> > ### Author Response · Authors · 2025-11-26
> > **Response to Reviewer comments**
> >
> > Dear reviewers and Action Editors,
> >
> > Thank you very much for your support and feedback throughout the entire review process. I understand that the manuscript in the current form still needs some significant improvement and elimination of errors, specifically with regard to Theorem 3 and some of the assumptions I have made. To address your concerns, I have tried to make the following amendments to the manuscript:
> >
> > 1) I have eliminated the assumption the radius of the cover element $\epsilon$ is smaller than that of the radius of the function class being considered $r$ or vice-versa. Moreover, I am now making no assumptions on the ratio $r/\epsilon$ whatsover and derived Lemma A2 and proof of Theorem 3 based on these changes.
> > 2) With regards to the assumptions I make for deriving the growth rates, I explicitly mention that I consider the spectral norm of each layer i to be 1 ($|W_i|_2=1$ for all i). This will make the Jacobian term  $B^{jac}$ equal to 1 and allow a fairer comparison with respect to the remaining terms.
> > 3) Lastly, I would request the reviewers and action editors to kindly review the updated proof of Theorem 2 based on the PAC-Bayesian framework and make a decision based on the validity of the proof. I did not receive any comments from reviewer 3h1F regarding this and was looking forward to it.
> >
> > I know that changes closer to the deadline are not recommended and may not be considered, but I tried my best to address some of the concerns that the reviewers and Action Editors had. I apologize if it causes any inconvenience, and I can revert to the previous version of the manuscript if required. However, I would request the reviewers to consider this revision if possible.
> >
> > I thank everyone once again for your feedback and comments, and looking forward to some final comments and a decision regarding the manuscript.

---

### Decision · Action_Editor_CpVn · 2025-12-09

**Recommendation:** Reject

**Additional Comments:**

The paper  aims to prove bounds improving on [4,5] for fully-connected neural networks. Theorem 2 claims to remove a factor of the depth $D$ in the main spectrally normalized result from [4,5], whilst Theorem 3 claims to achieve a bound with logarithmic dependence on depth for neural networks under the assumption that the spectral norms of the matrices are upper bounded by $1$.


The submission and proofs have been carefully reviewed by Reviewer 3h1F and by myself, uncovering **many serious errors**. A non exhaustive list of the most serious problems is as follows:

1. The proof of Theorem 2 (which claims to improve the bound of [4,5] by a factor of the depth) originally **failed to sum the contributions** of each layer's to the final granularity/perturbation $\epsilon$. The proof was almost exactly the same as in [5], except for the introduction of errors, which account for the claimed improvements. The error was caught by Reviewer 3h1F. Independently, there were **also errors** in restatements of Dudley's entropy theorem and **similar classic results** which were caught by both reviewer 3h1F and myself.

1.2 The first revision attempted to fix the problem above by rescaling the granularities of each layer by a factor of $1/D$ without taking into account the effect on the margin, providing a second, **also incorrect proof of the same result**. The error was caught by Reviewer 3h1F.

1.3 The final revision attempts to prove the same result with a completely different proof strategy relying on PAC Bayes analysis [4]: the idea is to reproduce the proof of the main result in [4] whilst using a different variance for the prior and the posterior distribution, which affects the calculation of the divergence. Reviewer 3h1F claimed the new proof was correct, though they admitted this was a low confidence statement as they are less familiar with the PAC Bayes aspects of the proofs. I have read the proof and can confirm it is **still incorrect**: the dimensionality of the space of weights is incorectly assumed to be $p$ instead of $p^2$ (if looking at a single layer) or $p^2D$ if looking at all layers simultaneously).


2. The parameter-counting argument in Lemma A2 contained errors regarding edge cases (as stated, it could involve an upper bound smaller than 1 for the covering number). This is a comparatively more minor issue as it is fixable.  I have discovered the error and mentioned it to the authors.

2.1 Subsequently, the authors uploaded a new version attempting to fix this error, but **still incorrectly**.

2.2 On the third try, this minor error was fixed.


3. Theorem 3 claims that the sample complexity of Neural Networks scales logarithmically with depth. As I mentioned in more details in my comment to the authors (https://openreview.net/forum?id=aSv29Xh81q&noteId=JMlriNvtnd), this is a very strong claim which, at least at face value, contradicts common knowledge and VC lower bounds. Admittedly, the authors made it clear that they are assuming $\|W_i\|\leq 1$, however, this condition isn't enough either. Admittedly, if $\|W_i\|<c$ for some $c<1$, it is credible that virtually any result could be obtained if one doesn't change the margin definition (because the maximum achievable margin decays exponentially with depth, making the bound meaningless). However, even assuming such a prohibitively strong condition, the proof given is **also incorrect** in all versions submitted so far. Indeed, the dependency on depth disappears into the logarithmic factors on page 27 where the authors incorrectly simplify equation (59). There are many problems here: first of all, the authors assume that $\frac{b_i}{\|W_i\| pr2\sqrt{2}}$ is "much smaller than 1" to use the inequality $\prod (1+x_i)\leq \sum(1+x_i)$ for small $x$. Apart from the fact the condition on $x_i$ is not explicitly verified, the inequality itself clearly doesn't hold for $x_i\simeq 1$, so the conditions are not clearly stated. The authors are not using the fact that $\lim_{n\rightarrow \infty} (1+\frac{a}{n})^{n}=e^a$ (for $n=pr$) but instead articulating their argument around the fact that the quantity inside the brackets is "approximately closer to zero provided that $p$ is large enough".



The comparison to the related works is also incomplete. For instance, the  authors claim to be able to prove bounds of order $O(\frac{\sqrt{Dpr}}{\sqrt{m}})$ where $D$ is the depth, $p$ is the width and $r$ is the rank. During the discussion, it transpired that this makes the assumption that $\|W_i\|\=1$ for all $I$. Under this condition, the result (with logarithmic factors, which will be present if the proof is made correct) is covered in both [3] and [6].


In spite of all the above, I believe that modifying the proof of Theorem 2 to make it correct **may possibly yield a result which would be technically original, although extremely incremental**: in future research in this direction, it seems realistic to obtain first a sample complexity bound of the form $O(p\log(pD)\prod_\{i=1\}^D $ $\|W_i\|^2 \sum_\{i=1\}^D $ $\frac{\|W_i\|^2_\{FR\}}{\|W_i\|^2}+ Dp^2\log(D))$ (note that the second term should indeed read $Dp^2log(D)$, not $Dp\log(D)$). If correct, the improvement over [6,3] would be subtle and mostly about smaller log factors (because a bound of $\tilde{O}(Dp^2)$ is known if $\|W_i\|=1$ for all $i$) and a shorter proof. Different variance ratios might yield slightly different results. The authors can also explore adding a low-rank prior. However, such results are probably **not enough on their own** to justify publication, so there is no guarantee of acceptance and further research/literature review/experimental evaluation would be required. The authors are advised to have their work checked by other researchers *before* any attempt at resubmission, whether here or to another journal.





References:

[1] Renjie Liao. Notes on rademacher complexity, 2020

[2] Roman Vershynin. High-dimensional probability. 2018

[3] Xingguo Li, Junwei Lu, Zhaoran Wang, Jarvis Haupt, and Tuo Zhao. On tighter generalization bound for deep neural networks: Cnns, resnets, and beyond, 2018 (preprint)

[4] Behnam Neyshabur, Srinadh Bhojanapalli, and Nathan Srebro. A pac-bayesian approach to spectrally- normalized margin bounds for neural networks, 2018 (ICLR)

[5] Bartlett et al. Nearly-tight vc- dimension and pseudodimension bounds for piecewise linear neural networks. JMLR 19.

[6] Ledent et al. Generalization Bounds for Rank-sparse Neural Networks. NeurIPS 2025.

[7] Graf. et al. On Measuring Excess Capacity in Neural Networks. NeurIPS 2022.

**Audience:**

No

**Audience Explanation:**

The topic is well within TMLR scope, but the results in their current form don't make sense.

**Claims And Evidence:**

No

**Claims Explanation:**

The paper contains an exceptionally high number of obvious, fatal mathematical errors. Nearly all the results are completely incorrect.

**Resubmission Of Major Revision:**

The authors may consider submitting a major revision at a later time.